# Regulation of branching dynamics by axon-intrinsic asymmetries in Tyrosine Kinase Receptor signaling

Marlen Zschätzsch[1,2,3], Carlos Oliva[1,2], Marion Langen[1,2,4†], Natalie De Geest[1,2], Mehmet Neset Özel[5], W Ryan Williamson[5‡], William C Lemon[6], Alessia Soldano[1,2], Sebastian Munck[1,7], P Robin Hiesinger[5], Natalia Sanchez-Soriano[8*], Bassem A Hassan[1,2,3,4,6*]

[1]Center for the Biology of Disease, Vlaams Instituut voor Biotechnologie, Leuven, Belgium; [2]Center of Human Genetics, University of Leuven School of Medicine, Leuven, Belgium; [3]Program in Molecular and Developmental Genetics, Doctoral School for Biomedical Sciences, University of Leuven Group Biomedicine, Leuven, Belgium; [4]Program in Molecular and Cognitive Neuroscience, Doctoral School for Biomedical Sciences, University of Leuven Group Biomedicine, Leuven, Belgium; [5]Department of Physiology and Green Center for Systems Biology, University of Texas Southwestern Medical Center, Dallas, United States; [6]Janelia Farm Research Campus, Howard Hughes Medical Institute, Ashburn, United States; [7]Bio Imaging Core, Vlaams Instituut voor Biotechnologie (VIB), Leuven, Belgium; [8]Department of Cellular and Molecular Physiology, Institute of Translational Medicine, University of Liverpool, Liverpool, United Kingdom

**\*For correspondence:**
N.sanchez-soriano@liv.ac.uk
(NS-S); bh@kuleuven.be (BAH)

**Present address:** †Department of Physiology, University of Texas Southwestern Medical Center, Dallas, United States; ‡Janelia Farm Research Campus, Howard Hughes Medical Institute, Ashburn, VA 20147, USA

**Competing interests:** The authors declare that no competing interests exist.

**Reviewing editor**: Franck Polleux, Columbia University, United States

**Abstract** Axonal branching allows a neuron to connect to several targets, increasing neuronal circuit complexity. While axonal branching is well described, the mechanisms that control it remain largely unknown. We find that in the *Drosophila* CNS branches develop through a process of excessive growth followed by pruning. In vivo high-resolution live imaging of developing brains as well as loss and gain of function experiments show that activation of Epidermal Growth Factor Receptor (EGFR) is necessary for branch dynamics and the final branching pattern. Live imaging also reveals that intrinsic asymmetry in EGFR localization regulates the balance between dynamic and static filopodia. Elimination of signaling asymmetry by either loss or gain of EGFR function results in reduced dynamics leading to excessive branch formation. In summary, we propose that the dynamic process of axon branch development is mediated by differential local distribution of signaling receptors.

## Introduction

The establishment of functional neuronal networks relies on the correct incorporation of a neuron into a developing circuit. An extended neurite network enables a single neuron to process information from multiple input cells and to relay that information to a wide range of targets. Neurite formation during development is a dynamic process and therefore tight regulation seems necessary to achieve connection specificity. At earlier steps of circuit formation, axon guidance, an intensively investigated process, combines intrinsic factors and extracellular cues to form a trajectory towards the general target area (*Williams et al., 2003*; *Schnorrer and Dickson, 2004*; *Kolodkin and Tessier-Lavigne, 2011*; *Pappu et al., 2011*). Subsequently, the formation of precise axonal connections within the target area relies on the development of the correct number of axonal branches. Currently, the

**eLife digest** In the human brain, 100 billion neurons form 100 trillion connections. Each neuron consists of a cell body with numerous small branch-like projections known as dendrites (from the Greek word for 'tree'), plus a long cable-like structure called the axon. Neurons receive electrical inputs from neighboring cells via their dendrites, and then relay these signals onto other cells in their network via their axons.

The development of the brain relies on new neurons integrating successfully into existing networks. Axon branching helps with this by enabling a single neuron to establish connections with several cells, but it is unclear how individual neurons decide when and where to form branches. Now, Zschätzsch et al. have revealed the mechanism behind this process in the fruit fly, *Drosophila*.

Mutant flies that lack a protein called EGFR produce abnormal numbers of axon branches, suggesting that this molecule regulates branch formation. Indeed in fruit flies, just as in mammals, the developing brain initially produces excessive numbers of branches, which are subsequently pruned to leave only those that have formed appropriate connections. In *Drosophila*, an uneven distribution of EGFR between branches belonging to the same axon acts as a signal to regulate this pruning process.

To examine this mechanism in more detail, high-resolution four-dimensional imaging was used to study brains that had been removed from *Drosophila* pupae and kept alive in special culture chambers. Axon branching and loss could now be followed in real time, and were found to occur more slowly in brains that lacked EGFR. The receptor controlled the branching of axons by influencing the distribution of another protein called actin, which is a key component of the internal skeleton that gives cells their structure.

In addition to providing new insights into a fundamental aspect of brain development, the work of Zschätzsch et al. also highlights the importance of stochastic events in shaping the network of connections within the developing brain. These findings may well be relevant to ongoing efforts to map the human brain 'connectome'.

mechanisms regulating axonal branch number and accuracy are largely unknown and subject to much debate.

In mammals, a common mechanism to regulate axon branch number is excessive axonal outgrowth and exuberant branch formation during development followed by a refinement process called pruning (*Low and Cheng, 2006*). Pruning encompasses the removal of relatively short axon terminals and branch arbors innervating a common target area as seen in the mouse peripheral and central nervous systems (*Sanes and Lichtman, 1999*; *Hashimoto et al., 2009*). In addition, long axon collaterals innervating distant target areas occurring for example in corticospinal tract (CST) axons of layer V neurons can be eliminated (*Weimann et al., 1999*). Removal of short redundant or inappropriate branches occurs typically via retraction of short branches whereas longer tracts are eliminated primarily by degeneration (*Luo and O'Leary, 2005*). A process involving features of both pruning mechanisms, termed axosome shedding, has been observed in mammals (*Bishop et al., 2004*).

An important question is how branch refinement is regulated. For a long time activity-dependent mechanisms were thought to be the major factor underlying regulation of pruning in the mammalian system (*McLaughlin et al., 2003*; *Yu et al., 2004*; *Huberman et al., 2006*; *Hashimoto et al., 2009*). However, several studies in various vertebrate systems suggest that this may not be universally true (*Crowley and Katz, 2000*; *Bagri et al., 2003*; *Pfeiffenberger et al., 2006*; *Cang et al., 2008*; *Sun et al., 2011*; *Wei et al., 2011*). Thus, although there is ample description of axonal branch refinement in vertebrate systems, much remains to be elucidated about the mechanisms underlying them.

In *Drosophila* deterministic genetic programs are thought to account for the stereotypic development of the vast majority of neuronal connections (*Jefferis et al., 2001*; *Hiesinger et al., 2006*). Nevertheless, a specialized form of pruning also occurs in *Drosophila*, namely the widely studied remodeling of insect networks during metamorphosis. In holometabolous insects, like the fruit fly, many cells need to accommodate two distinct morphological and behavioral states within a lifetime. In the nervous system neuronal arbors have to remodel extensively to allow the reiterative use of larval neuronal populations to form adult circuits. Interestingly, the molting hormone

Ecdysone is not only necessary for body transformation but also for the regulation of remodeling events in the nervous system (*Truman, 1990*). This system resembles partially the emergence of an adult network from initial projections as seen in vertebrates in the visual and motor cortex (*O'Leary and Koester, 1993*).

In this study, we focus on axonal branch refinement of the dorsal cluster neurons (DCNs) in the central nervous system (CNS) of *Drosophila* (*Hassan et al., 2000*). DCNs form only adult-specific neuronal projections and therefore unlike sensory neurons (*Williams et al., 2006*) and mushroom body neurons (*Boulanger et al., 2011*), DCN axons are not remodeled during metamorphosis. DCN axons innervate the optic lobes via an initial phase of long-range axonal growth and retraction steps, followed by the establishment of a stereotypic number of axonal branches by an unknown mechanism. In this work, we first describe that this wiring pattern is achieved through initially excessive axonal branch growth followed by refinement during brain development. Next, we show that the refinement process is regulated through local activation of EGFR signaling in part by EGF-secreting sensory axons. We find that EGFR shows intrinsic differential distribution between individual developing DCN axonal branches and that the appropriate level of signaling is required for proper axonal branching. Mechanistically, we find that, in this context, the EGFR acts via regulating actin cytoskeleton dynamics, and not the canonical mitogen activated kinase (MAPK) pathway. Finally, high-resolution 4D live imaging of pupal brain explants shows that inhibition of EGFR signaling causes a dramatic reduction in axonal branch dynamics leading to the failure of axonal branch pruning.

## Results

### Dorsal cluster neurons as a model to study axonal branch formation

The dorsal cluster neurons (DCNs) establish a complex neurite network in the *Drosophila* adult optic lobes. A small subset of neurons from this cluster extend their axons in the outer part of the optic lobe, the medulla (Me) (*Srahna et al., 2006*; *Langen et al., 2013*), where they form a stereotypic pattern of axonal branches (*Figure 1A,B*). This pattern can be readily visualized using the *ato-Gal4* driver in combination with a UAS-driven marker of choice such as CD8-GFP. Flip-out single cell clones (*Wong et al., 2002*) reveal the branch pattern of an individual axon derived from a single neuron of the 12 medulla innervating DCNs (*Figure 1C*). False color labeling and tracing (*Longair et al., 2011*) of single DCN Me axons and their branches (*Figure 1D*) reveals that each axon generates 6–8 primary branches, with a mean of 7 branches. This stereotypic pattern is achieved by hot spots of branches extending in dorsal and ventral direction from each main axon shaft. The first main branch point is located at the border between lobula and Me with one or two branches. The next major branch point with often two branches is situated in Me layers M7–M8 and in this location branches from distinct neighboring axons are often in close contact forming a grid-like pattern. The terminal set of up to four branches is distributed over the M1–M3 layers and is more often intermingled with neighboring axon branches. In between the two most distal branch points intermediate branches occur occasionally. DCN branches never extend beyond the Me neuropil.

### EGFR signaling regulates axon branch formation

We carried out a targeted screen using loss and gain of function transgenes for signal transduction and axon guidance receptors to identify pathways that might regulate axon branch development. We noted excessive branching in the adult DCNs using a dominant-negative construct of the EGFR. To validate these findings we first analyzed flies carrying a viable hypomorphic loss of function mutation for the receptor (EGFR$^{T1}$). In this genetic background DCN axons show short ectopic branches (*Figure 2A*) highlighted using the tracing tool (*Figure 2A'*). Since the proper development of the optic lobes depends on EGFR signaling (*Huang et al., 1998*), reduced EGFR signaling might indirectly influence DCN axon formation and morphology. To investigate whether the EGFR is required in the DCNs for axonal branch refinement, we sought to generate DCN MARCM EGFR-null clones (*Lee and Luo, 1999*), whereby EGFR function is removed at the time of neuronal birth. We obtained very few clones, suggesting that the EGFR may be required early during development for cell viability. The clones we did obtain showed ectopic branching defects, but also severe axon targeting phenotypes, suggesting that the EGFR is required early in DCN development and precluding further analysis of these clones (*Figure 2—figure supplement 1*). To avoid these early defects, we used the *ato-Gal4* driver, which is expressed in postmitotic DCNs after the initiation of axonal outgrowth (*Srahna et al.,*

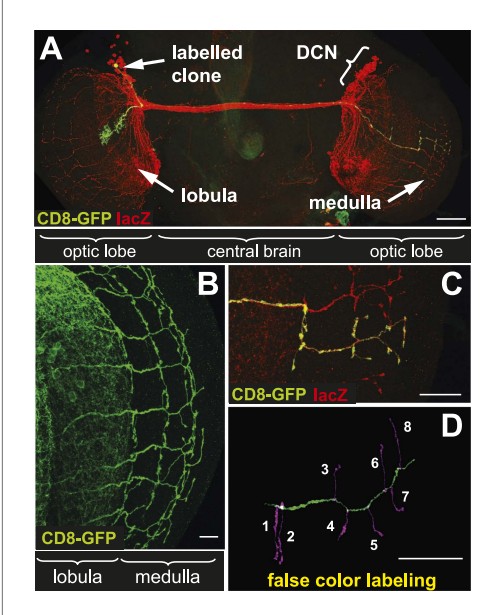

**Figure 1**. The axonal network of medulla dorsal cluster neurons (DCNs) in the adult central nervous system of Drosophila. (**A**) Dorsal cluster neurons, labeled with lacZ (red) using the atoGal4-14a driver, with its dendritic and axonal projections in the optic lobes of the CNS. Using the FLP-out system, an individual neuron is labeled with mCD8-GFP (green) within the background of the entire cluster. AtoGal4-14a is used in all the following experiments except when stated otherwise. (**B**) DCN axons, labeled with mCD8-GFP, form a stereotypic pattern of axonal branches within the medulla (Me) of the adult optic lobe. (**C**) Using the FLP-out system the axon and branches of an individual neuron are labeled with mCD8-GFP (green) within the background of the entire cluster labeled with lacZ (red). (**D**) False color labeling of one Me DCN axon with its main shaft (green) and branches (magenta) using a tracer tool. The scale bars represent 100 μm in (**A**) and 20 μm in (**B–D**).

2006; *Zheng et al., 2006*; *Langen et al., 2013*), to express two different dominant negative alleles of the EGFR (uas-EGFR^DN-A, *Freeman, 1996*; uas-EGFR^DN-B, *Buff et al., 1998*) and *EGFR^RNAi* (uas-EGFR^RNAi, VDRC107130). In all three cases the DCNs show a significant increase of axon branches in the adult CNS. Compared to an average of 7 primary branches under wild type conditions, we observed a significant increase to 10.5 primary branches per axon in EGFR^DN-A expressing DCNs (*Figure 2B,B',E*). Single cell clones in wild type (*Figure 2F*) and EGFR^DN–A background (*Figure 2G*) show the branch increase on single cell level. Expression of the second, weaker, EGFR^DN allele (*Urban et al., 2004*) (EGFR^DN-B) resulted in an increase to an average of 8.3 branches per axon (*Figure 2C,C',E*), and EGFR knock-down with RNAi leads to a similar increase to 8.5 branches per axon (*Figure 2D,D',E*). In the case of the EGFR^DN-A axonal branches appear thin and spike-like suggesting that they are immature. Interestingly, inhibition of EGFR results not only in an increase of the average branch number, but also increases the variability in the branch numbers between individual axons (*Figure 2—figure supplement 3*), even within the same individual brain, suggesting that EGFR signaling may regulate the accuracy and robustness of the branching process.

Activation of the EGFR requires binding to its EGF ligands. To confirm that EGFR signaling regulates DCN axon branching, we first tested adult hypomorphic mutants for the EGFR ligand Spitz (Spi) whose role in optic lobe development is well described (reviewed in *Salecker et al., 1998*). Reduction of Spi activity results in ectopic short branches indistinguishable from those seen in EGFR hypomorphic mutants (*Figure 2—figure supplement 2A,A'*; compare to *Figure 2A,A'*). To determine the source of the EGF signal that regulates DCN branch refinement, we considered two possibilities. First, DCN axons themselves might release an activating ligand to initiate an autocrine signaling mechanism, as seen in the p75-TNR axoaxonal competition of mouse and rat sympathetic axons innervating the eye (*Singh et al., 2008*). Second, neurons in the target neuropil might release EGF to regulate branch refinement. A subset of retinal photoreceptors known as R8 and R7 have axon terminals that innervate the medulla. Photoreceptors are known to secrete Spi to initiate a number of EGFR-dependent events in the developing optic lobes (*Huang and Kunes, 1998*; *Huang et al., 1998*; *Yogev et al., 2010*). We analyzed the coincidence of innervation of the medulla by R7 and R8 photoreceptor axons using the photoreceptor specific marker mAb 24B10 (*Fujita et al., 1982*). Overlap between DCN and R7/8 axons can be seen at different times during brain development and in the adult brain (*Figure 2—figure supplement 2B–D*).

To distinguish between the two models, we used *Spi^RNAi* driven by either ato-Gal4 (DCNs) or GMR-Gal4 (photoreceptors) to down regulate *Spi* expression in the DCNs or photoreceptors, respectively. To visualize DCN branch formation while down regulating *Spi* specifically in the photoreceptors, we used the Gal4-independent LexA-based binary expression system (*Lai and Lee, 2006*). Specifically, we took advantage of the *ato^LexA* IMAGO (*Choi et al., 2009*) knock-in allele we

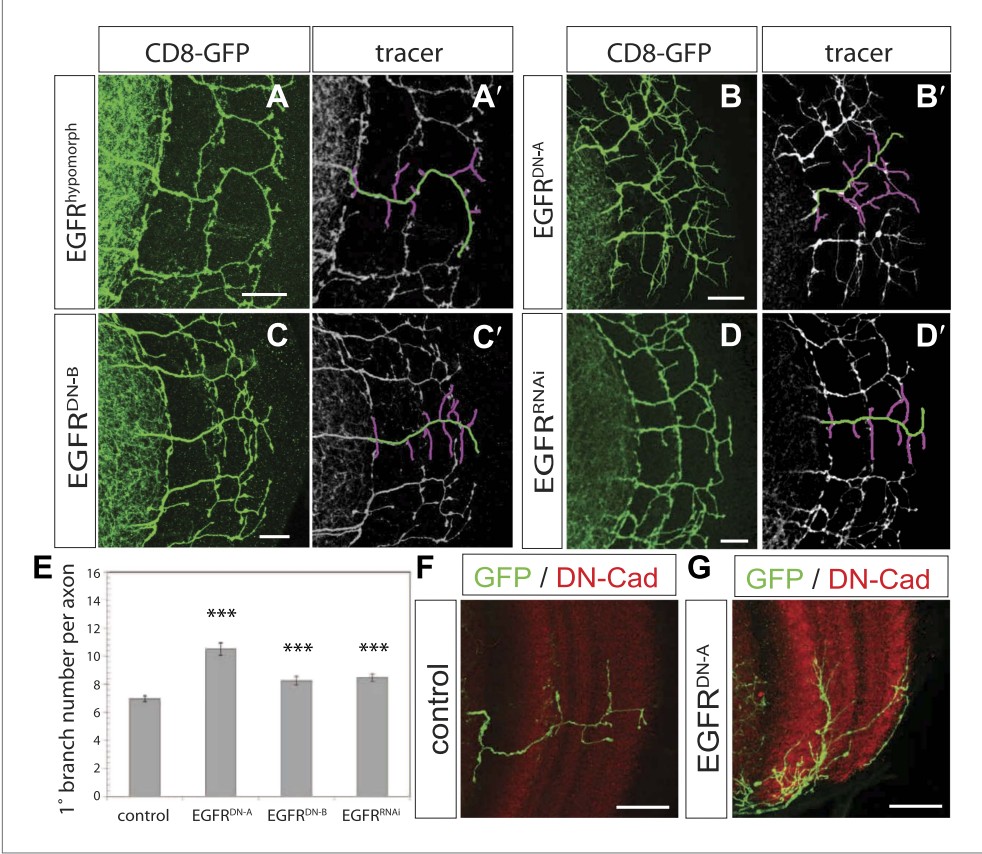

**Figure 2**. EGF-receptor downregulation in the DCNs results in excessive axonal branches in the adult. (**A**) The homozygous hypomorphic allele EGFR$^{T1}$ shows additional, short branches. (**B**–**D**) Downregulation of the EGFR specifically in the DCNs results in an increase of adult branches via overexpression of (**B**) a dominant-negative form **A** (UAS-EGFR$^{DN-A}$), (**C**) a dominant-negative form **B** (UAS-EGFR$^{DN-B}$) and (**D**) a RNAi against EGFR (UAS-EGFR$^{RNAi}$). (**A′**–**D′**) Visualization of branches (purple) along a single main axon shaft (green) using the tracing tool reveals excessive branches of the aforementioned genotypes in (**A**–**D**). (**E**) Quantification of adult primary branch numbers per axon for the genotypes shown in (**B**–**D**) shows significant increase of branches. Control 6.96 ± 1.34 (n = 60), EGFR$^{DN-A}$ 10.5 ± 2.5 (n = 45, p<0.001) EGFR$^{DN-B}$ 8.3 ± 1.46 (n = 40, p<0.001), *EGFR$^{RNAi}$* 8.5 ± 1.09 (n = 40, p<0.001). (**F**–**G**) Adult Drosophila brain in which the neuropil is marked with DN-Cad (red). Flip out DCN clones are generated in control (**F**) and EGFR$^{DN-A}$ (**G**) background. Error bars represent SEM. Non-parametric ANOVA Kruskal–Wallis test. \*\*\*p<0.001. The scale bars represent 20 µm.

The following figure supplements are available for figure 2:

**Figure supplement 1**. EGFR$^{null}$ MARCM clones show early branch growth defects.

**Figure supplement 2**. Spi release from photoreceptor axons regulates DCN axon branch pruning.

**Figure supplement 3**. Distribution of axon branch numbers in control and EGFR-DN flies.

recently generated (*Langen et al., 2013*) and used it to drive LexAop-GFP expression in DCNs. Whereas we find no significant difference in branch number upon knock-down of *Spi* in the DCNs (*Figure 2—figure supplement 2F,G,J*), *Spi* knock-down in photoreceptors causes a significant increase in DCN branches (*Figure 2—figure supplement 2H,I,J*). In addition, to Spi release from photoreceptors, we observed cells expressing a reporter for the EGFR ligand Vein in close proximity to DCN axons (*Figure 2—figure supplement 2E*), suggesting a second source of EGF within the brain. Taken together, these results show that EGFR signaling regulates DCN axonal branch development.

## EGFR is required for developmental axon branch pruning

In theory, the adult branching pattern of DCN medulla axons can be established via one of at least two distinct mechanisms during development. On the one hand, accurate target innervation might proceed via the direct formation of the correct number of branches. Alternatively, the specificity of axonal branching might be the result of initial excessive outgrowth and exuberant branch formation during development followed by a refinement process to eliminate the majority of branches, as in refinement observed in mammalian visual map formation (*Feldheim and O'Leary, 2010*), for example.

To distinguish between these two models, we characterized branching of wild type DCN axons at different time points after puparium formation (APF) during brain development. Between 36 hr and 54 hr APF DCN axons form extensive branches at multiple positions along the growing axon (*Figure 3A–C*). Between 60 hr and 72 hr APF pruning begins to be evident (*Figure 3D–F'*). At 84 hr APF, the eventual adult branch pattern of 6–8 branches is apparent (*Figure 3G*) and little or no further pruning appears to occur beyond that point (*Figure 3H,I*). This developmental pattern is not an artifact of the expression of the membrane bound marker CD8-GFP, as two other intracellular axonal markers (nSyb-GFP and Syt-GFP) yield the same results (*Figure 3—figure supplement 1*).

Excessive axonal branches in EGFR mutant adults may be the result either of increased branch growth or of failure of branch pruning. To distinguish between these two possibilities, we analyzed DCNs expressing EGFR$^{DN}$ during pupal development. Between 36 hr and 48 hr APF axon branching at the second branch point is similar to wild type (*Figure 3J,K*). An initial difference in branch phenotype can be observed at 60 hr APF and subsequently at 72 hr APF, the typical refinement seen in wild type is largely absent in the EGFR$^{DN}$ background (*Figure 3L–M'*). The failure to prune is evident at 84 hr and 96 hr APF (*Figure 3N,O*) where DCN axons show excessive axonal branches. To rule out developmental delay as a cause we examined 2-day vs 18-day-old EGFR$^{DN}$ flies. These flies are indistinguishable from 96 hr APF EGFR$^{DN}$ flies indicating no further branch refinement (*Figure 3—figure supplement 2*). Finally, we quantified axonal branch pruning at 48 hr and 72 hr APF by counting the number of branch end-points at these two time points in wildtype and EGFR$^{DN}$ flies, respectively. While there is no significant difference between the two genotypes at 48 hr APF, quantification at 72 hr APF confirms the increased amount of branches in the EGFR$^{DN}$ background compared to wild type (*Figure 3P*). In addition, the significant decrease in branch number seen in wild type axons between 48 hr and 72 hr is not observed in the EGFR$^{DN}$ axons (*Figure 3P*). In summary, these data show that EGFR signaling is required to generate the correct number of axonal branches through the reduction of branch precursors formed during development (*Figure 3Q*).

## Asymmetry of EGFR localization regulates differential filopodial dynamics

To gain insight into the role of EGFR during axonal branching, we turned to primary embryonic *Drosophila* neuronal culture (*Prokop et al., 2011*; *Sanchez-Soriano et al., 2010*). After 2 days in culture wildtype *Drosophila* primary neurons sprout on average ~2.5 primary axonal branches, whereas neurons expressing EGFR$^{DN}$ show a significant increase in branch number (*Figure 4A–C*), suggesting that regulation of axonal branching by the EGFR is a process intrinsic to neurons and common to different neuronal subtypes. Axonal branches develop from dynamic filopodia that gets stabilized during the axonal branching process. We quantified the dynamics of filopodia under WT and EGFR loss of function conditions. We find that in growing wildtype neurons less than 10% of filopodia are static during the imaging time window of 3 min. In contrast, EGFR$^{DN}$ neurons have a significant increase in the percentage of static filopodia to ~30% (*Figure 4D–F*). An increase in static filopodia may suggest that more of the transient protrusions are stabilized into branches. An indication of the maturation of filopodia into branches is the invasion of microtubules into axonal filopodia (*Gallo, 2011*). Accordingly, we find that EGFR$^{DN}$ induces an increase in microtubules invading axonal filopodia (*Figure 4—figure supplement 1*). These data suggest that EGFR signaling regulates branch formation by controlling the dynamics of immature protrusions. To examine the localization of the EGFR in primary neurons, we cultured neurons from animals expressing C-terminally GFP-tagged EGFR (UAS-EGFR$^{GFP}$) and performed live imaging experiments. We find that the EGFR is dynamically transported into and out of axonal branches and their filopodia (*Figure 5A*; *Video 1*, *Video 2*), with slightly, but significantly, higher levels in dynamic filopodia compared to static filopodia (*Figure 5A', A''*, *Figure 5—figure supplement 1*). In summary, our data indicate that EGFR is differentially localized to static vs dynamic filopodia and that its activity promotes dynamic filopodial behavior and consequent adjustment of branch number.

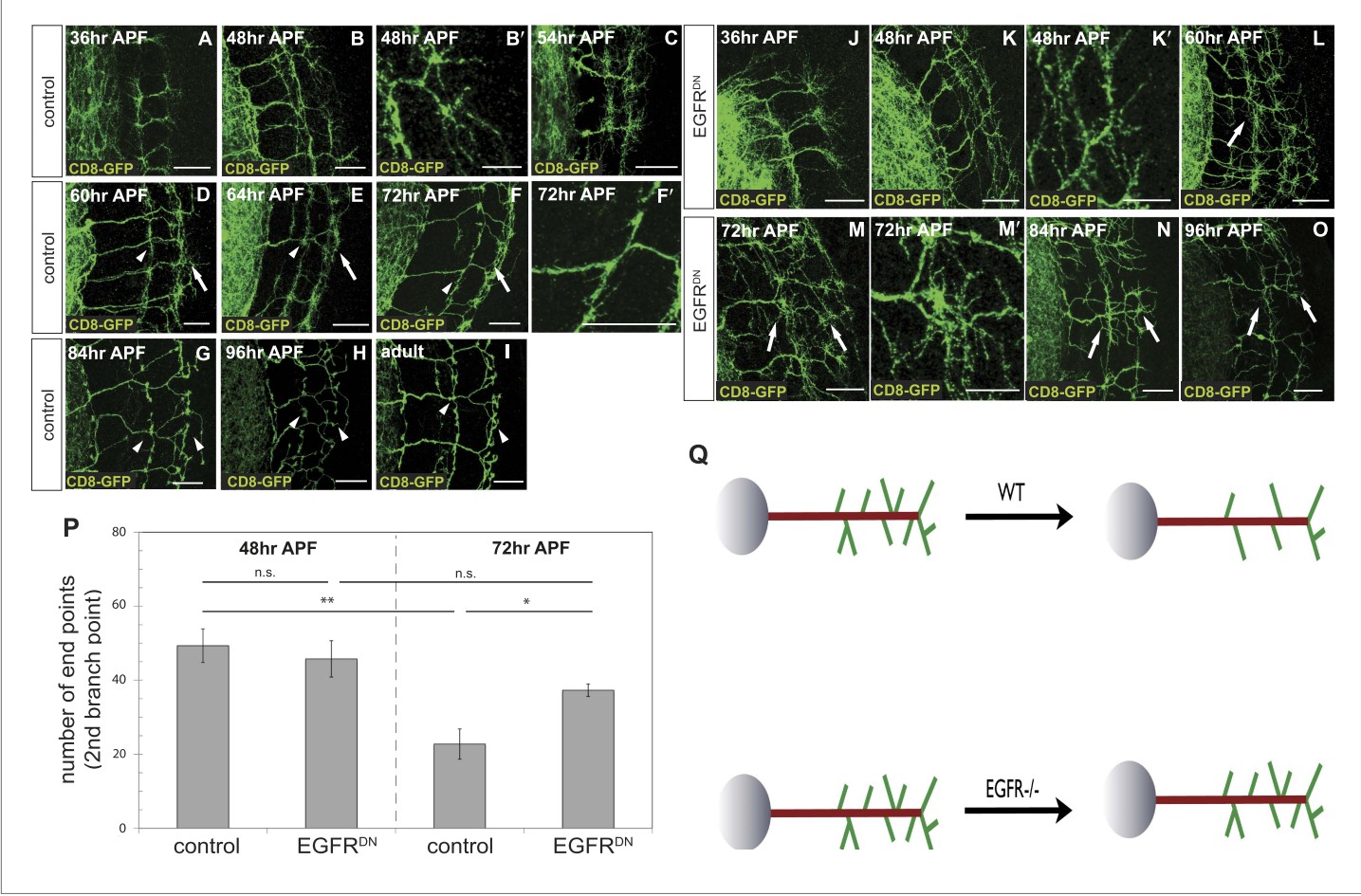

**Figure 3**. Loss of EGFR function impairs developmental axon branch pruning. (**A**–**I**) Axonal branch pattern at different pupal stages shows excessive branching at early to mid-pupal development. Successive refinement of exuberant branches can be observed between 60 hr and 96 hr (arrowhead, compare **D**–**H**). Branch morphology at (**A**) 36 hr APF, (**B** and **B'**) 48 hr APF, (**C**) 54 hr APF, (**D**) 60 hr APF, (**E**) 64 hr APF, (**F** and **F'**) 72 hr APF, (**G**) 84 hr APF, (**H**) 96 hr APF and (**I**) adult stage. High magnification of branches is shown in **B'** and **F'**. (**J**–**O**) Axonal branch pattern at different pupal stages of EGFR^DN express-ing DCNs shows excessive branching at early to mid-pupal time points similar to wild type. Impaired refinement of exuberant branches can be observed between 60 hr and 96 hr (arrow, compare **L**–**O**). Branch morphology at (**J**) 36 hr APF, (**K** and **K'**) 48 hr APF, (**L**) 60 hr APF, (**M** and **M'**) 72 hr APF, (**N**) 84 hr APF, (**O**) 96 hr APF. High magnification of branches is shown in **K'** and **M'**. (**P**) Quantification of branches at the second branch point at 48 hr and 72 hr APF comparing control and EGFR^DN using the Skeleton Analysis tool of ImageJ ('Materials and methods'). EGFR downregulation does not result in increased branches at 48 hr APF compared to control. Significant decrease of developmental branch numbers at 72 hr APF occurs due to refinement in control. No significant decrease in branch number was observed after EGFR downregulation between 48 hr and 72 hr APF. Compared to control more branches persist after EGFR downregulation at 72 hr APF. Control (48 hr APF) 49.33 ± 9.87 (n = 18), control (72 hr APF) 22.75 ± 9.1 (n = 18, p<0.01), EGFR^DN (48 hr APF) 45.77 ± 10.96 (n = 16), EGFR^DN (72 hr APF) 37.3 ± 3.83 (n = 14) (to control 72 hr APF, p<0.05). Error bars represent SEM. *t* test. *p<0.05; **p<0.01. The scale bars represent 20 µm except in **B'**, **K'** and **M'** with 10 µm. (**Q**) Schematic representation of the role of EGFR signaling in DCN axonal branch formation.

The following figure supplements are available for figure 3:

**Figure supplement 1**. DCN axon branches.

**Figure supplement 2**. Branch growth is not enhanced in aged EGFR^DN flies.

We wondered whether differential EGFR localization is itself dependent on EGFR signaling activity. To this end, we compared levels of EGFR^GFP in filopodia of control vs EGFR^DN neurons. We find that the difference in EGFR^GFP levels between dynamic and static filopodia drops dramatically upon inhibi-tion of EGFR signaling (***Figure 5B,B',B'',C***). EGFR signaling depends on receptor endocytosis upon ligand binding (***Haigler et al., 1979***). Interestingly, EGFR^GFP traffics actively all along the axonal shafts, branches and filopodia in cultured neurons (***Video 3***), and we find EGFR-GFP puncta partially

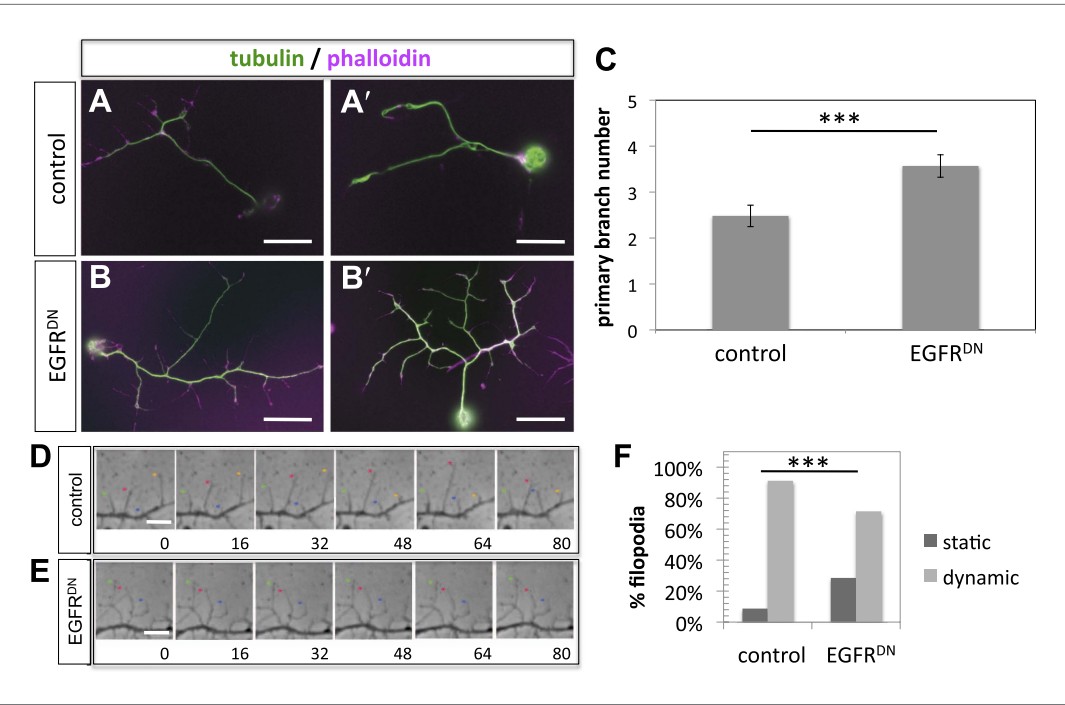

**Figure 4**. EGFR regulates filopodia dynamics in primary Drosophila neuronal cultures. (**A** and **B**) Branch formation in cultured primary Drosophila neurons (2 days). (**B–B'**) Overexpression of UAS-EGFR$^{DN}$ using the *sca-Gal4* driver results in an increase of branches when compared to (**A–A'**) wild type (control). For the visualization of branches, neurons were stained with anti-tubulin (green) and phalloidin (magenta). (**C**) Quantification of primary branch numbers per axon shows significant increase of branches in UAS-EGFR$^{DN}$ neurons (control: 2.48 ± 0.2 (n = 83); EGFR$^{DN}$: 3.57 ± 0.24; n = 74, p<0.001). (**D–E**) Still images from videos of (**D**) wild type and (**E**) UAS-EGFR$^{DN}$-expressing neurons. Overexpression of UAS-EGFR$^{DN}$ using the *sca-Gal4* driver results in a decrease of filopodia dynamics in primary Drosophila neurons cultured for 6–8 hr. Different filopodia are marked by colored arrows and can be followed over time. (**F**) Quantification of static vs dynamic (extensions and retractions) behaviors shows a significant distribution change between wild type vs EGFR$^{DN}$-expressing filopodia (control: static = 10, dynamic = 110; EGFR$^{DN}$: static = 41, dynamic = 86, p<0.001). Error bars represent SEM. Mann–Whitney test. ***p<0.001. The scale bars in (**A–B**) represent 10 μm and in (**D–E**) represent 3 μm.

The following figure supplements are available for figure 4:

**Figure supplement 1**. Increase in filopodia containing microtubules by expression of UAS-EGFR$^{DN}$.

co-localize with both Rab5 and Rab11, suggesting that EGFR is present on early and recycling endosomes (*Figure 5—figure supplement 2*). These data indicate that recycling might lead to differential EGFR localization in filopodia. Because the EGFR$^{DN}$ used here can still bind ligand but fails to signal, one interesting possibility is that these dominant negative receptors may titrate ligand away from the functional receptor and thus inhibit not only signaling, but also internalization. To test the putative role of endocytosis in receptor dynamics, we live-imaged EGFR$^{GFP}$ localization in filopodia before and after inhibition of endocytosis using the Dynamin inhibitor Dyngo (*Harper et al., 2011*). In untreated wild type neurons, EGFR$^{GFP}$ levels vary between individual filopodia and within each filopodium over time (*Figure 6A–C*). Upon inhibition of endocytosis, the overall levels of EGFR$^{GFP}$ in filopodia decrease and the fluctuation of EGFR$^{GFP}$ between and within filopodia is significantly reduced (*Figure 6A'–C*). This is accompanied by a dramatic reduction in filopodial dynamics (*Figure 6D*; *Video 4*), suggesting that receptor endocytosis and recycling regulates EGFR localization and dynamics in filopodia.

## EGFR signaling shows asymmetric localization and regulates differential filopodial dynamics in vivo

Next, we asked if the EGFR is differentially localized and regulates branching dynamics in vivo. *EGFR* transcription (*Schejter et al., 1986*) and function in the *Drosophila* developing and adult brain have

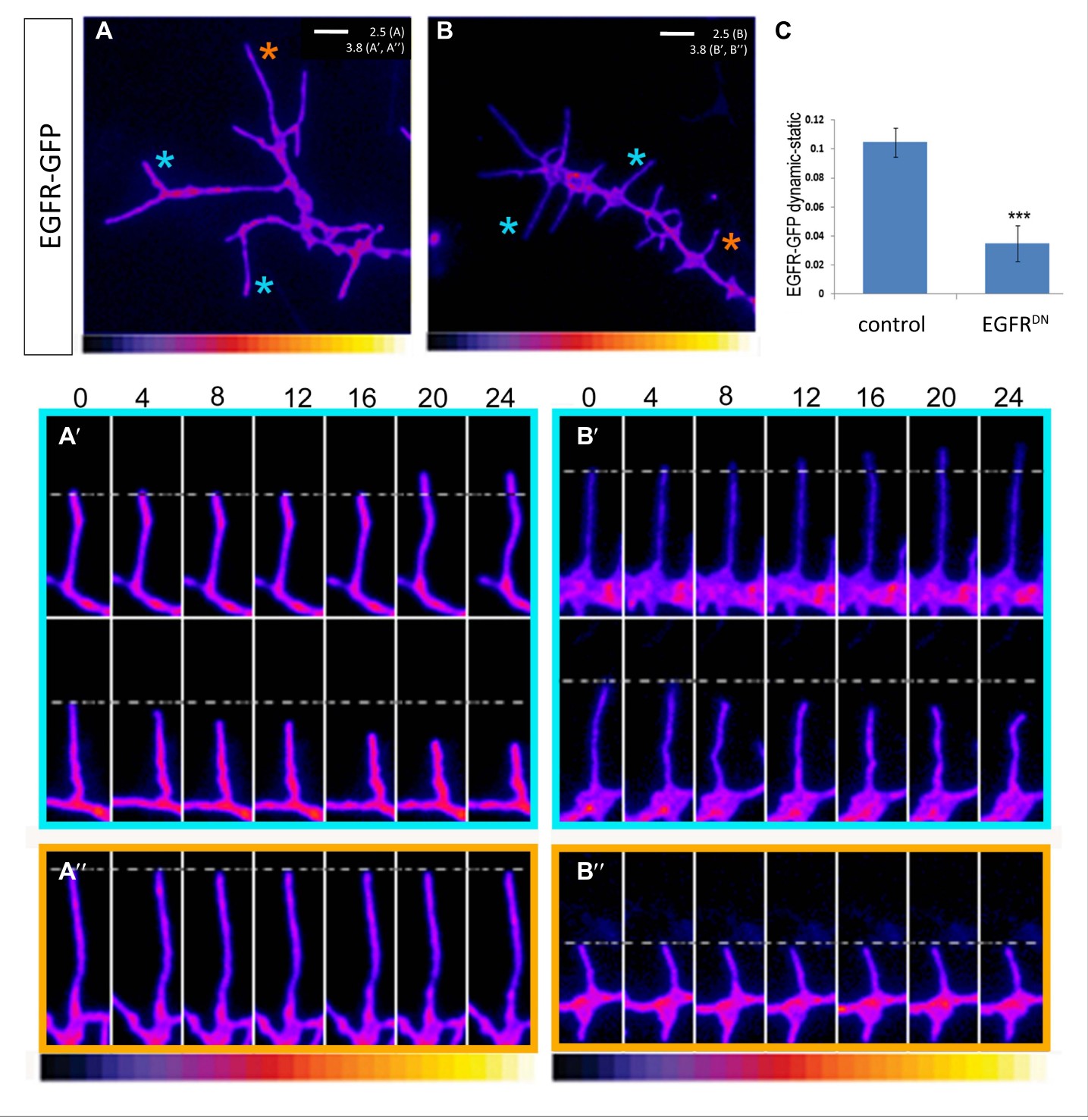

**Figure 5**. EGFR shows differential localization in filopodia of primary Drosophila neurons. (**A–B**) UAS-EGFR[GFP] expressed with elav-Gal4 in wild type (**A**) and EGFR[DN] (**B**) primary Drosophila neurons. False color image displaying a heat map of an EGFR[GFP]-expressing growth cone. EGFR[GFP] expression in dynamic (**A'** and **B'**) and static filopodia (**A"** and **B"**) is followed over time in wild type (**A'** and **A"**) and EGFR[DN] (**B'** and **B"**). **A'** and **B'** each shows one filopodia growing and one retracting (**C**). To quantify EGFR[GFP] intensity in static vs dynamic filopodia in the absence (control) or presence of EGFR[DN], we calculated the ratio of EGFR[GFP] in dynamic minus static filopodia (GFP maximal intensity of each dynamic phase minus the mean of GFP maximal intensity in static filopodia). The difference in EGFR[GFP] levels between dynamic filopodia and static filopodia are significantly reduced in the presence of
*Figure 5. Continued on next page*

*Figure 5. Continued*

EGFR[DN] (control dynamic-static: 0.1046 ± 0.009, n=216; EGFR[DN] dynamic-static: 0.0349 ± 0.0121, n=124, p<0.001). Error bars represent SEM. Mann–-Whitney test. ***p<0.001.

The following figure supplements are available for figure 5:

**Figure supplement 1**. Localization of EGFR in cultured neurons.

**Figure supplement 2**. Colocalization of EGFR with Rab11 and Rab5 in the growth cone.

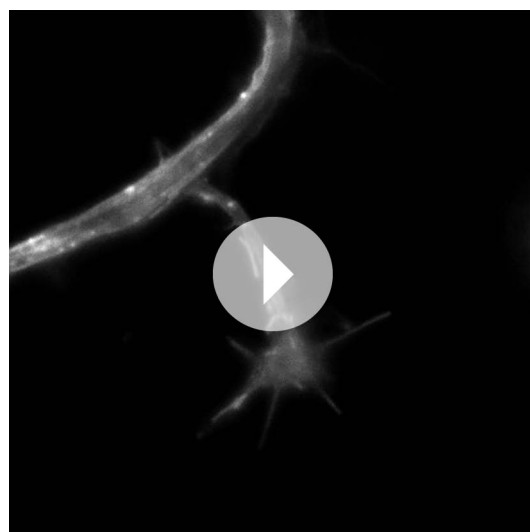

**Video 1**. EGFR-GFP cell culture filopodia. This video is related to *Figure 5*. Live imaging time-lapse video of axons from different primary neurons grown in culture for 4 days. UAS-EGFR[GFP] is expressed with elav-Gal4 driver. Images were collected every 4 s.

been documented, where it plays a role in neuronal survival (*Botella et al., 2003*) and sleep regulation (*Foltenyi et al., 2007*). However, attempts to detect the EGFR protein using immunohistochemistry have thus far failed, most likely due to very low expression levels. We attempted to circumvent this problem by generating a genomic rescue construct tagged at the C-terminal end with GFP, identical to the UAS-EGFR[GFP] used in our cell culture experiments. This construct rescues the embryonic lethality of EGFR null mutants to full adult viability with no visible defects. We examined the expression of genomic EGFR[GFP] during brain development and find that it is broadly expressed in the developing neuropil, especially the distal medulla (*Figure 7—figure supplement 1A*), suggesting that EGFR signaling may be generally involved in the regulation of CNS connectivity. Indeed, inhibition of EGFR activity in the lateral neurons ventral (LNv) also causes excessive axonal branching (*Figure 7—figure supplement 1B*). Unfortunately, expression levels of the genomic EGFR[GFP] transgene were too low to allow analysis at sub-cellular, single axon branch resolution. To examine subcellular EGFR distribution, we expressed UAS-EGFR[GFP] in the DCNs. In DCNs, EGFR[GFP] is detected in a punctate pattern in the cell bodies (*Figure 7A,B*, insets), along the axons and in axonal branches (*Figure 7—figure supplement 1C–E*). At ~56 hr APF, when extensive growth and pruning occur, EGFR[GFP] is unevenly distributed across different branches of the same axon (*Figure 7A–A′′′*) and we find no stereotypic pattern across different individual axons or individual brains (*Figure 7—figure supplement 1F*). At ~72 hr APF, after significant pruning has occurred, EGFR[GFP] is distributed more uniformly across the remaining unrefined branches (*Figure 7B–B′′′*). Note that both wildtype-untagged EGFR and EGFR[GFP] do not change the DCN branching pattern (*Figure 7—figure supplement 4*), hinting that asymmetric receptor signaling is governed by differential receptor distribution, rather than total receptor levels per se. In the LNv, EGFR[GFP] is expressed in cell bodies and low levels are present along the growing axons during development (*Figure 7—figure supplement 1G*). In contrast, in adult LNv UAS-EGFR[GFP] becomes restricted to neuronal soma (*Figure 7—figure supplement 1G′*). Thus, remarkably, even overexpressed EGFR[GFP] is present at relatively low levels and shows regulated developmental localization in different neuronal populations in vivo.

DCN branches develop and prune during pupal development when the brain is not easily accessible to live imaging. To overcome this limitation, we modified the protocol for long-term adult brain explant culture (*Ayaz et al., 2008*) to support long-term pupal brain culture. This protocol supports the morphologically normal development of *Drosophila* pupal brains (*Figure 7—figure supplement 2A–C*). We sought to probe the basis of the regulation of developmental branch pruning by the EGFR.

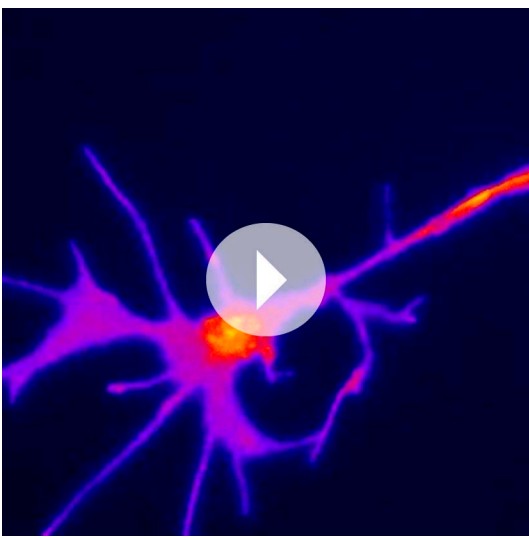

**Video 2**. EGFR-GFP cell culture filopodia. This video is related to **Figure 5**. Live imaging time-lapse video of axons from different primary neurons grown in culture for 4 days. UAS-EGFR$^{GFP}$ is expressed with elav-Gal4 driver. Images were collected every 4 s.

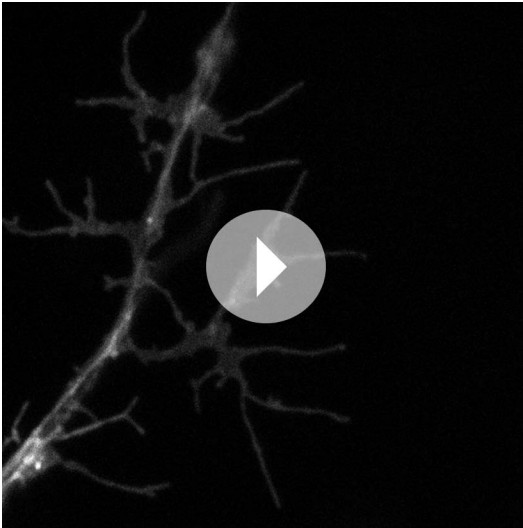

**Video 3**. EGFR-GFP cell culture filopodia. This video is related to **Figure 5**. Live imaging time-lapse video of axons from different primary neurons grown in culture for 4 days. UAS-EGFR$^{GFP}$ is expressed with elav-Gal4 driver. Images were collected every 4 s.

To this end, we performed high-resolution 4D live imaging to analyze real-time DCN axon branch formation by pairing the brain explant culture technique with resonant confocal microscopy of the cultured brains in a closed perfusion chamber (**Williamson and Hiesinger, 2010**). Imaging of developing wild type pupal brains (40 hr–60 hr APF) shows that wildtype DCN axon branches are dynamic (**Video 5**). Branch growth and removal occurs within minutes and can span up to 11.5 μm within 5 min with an average of 7.5 μm during this period (**Figure 7C$_1$–C$_3$,E**). Furthermore, wildtype branches behave differently from each other, as indicated by the spread of growth and retraction speeds of different branches (**Figure 7G**). In contrast, the growth dynamics in the EGFR$^{DN}$ expressing neurons are reduced in speed (**Video 6**). Growth and retraction processes of single branches are decreased to an average of 3 μm within 5 min and the dynamics show strikingly reduced variability between individual branches (**Figure 7D$_1$–D$_3$,F,G**). We wondered whether we could exploit the new developing brain culture system to ask whether EGFR is dynamically trafficked within DCN branches in vivo as these branches grow and retract. To this end, we generated flies expressing both EGFR-GFP and a red fluorescent protein (td-Tomato) in the DCNs. Live imaging (**Video 7**) of pupal brains from these animals and analysis of still images form these videos (**Figure 7—figure supplement 2D**) confirms that, like in primary neurons in culture, EGFR is trafficked dynamically as axons grows and retracts their branches in vivo; finally, it should be noted that EGFR$^{GFP}$ shows similar localization and activity to its wild type counterpart (**Figure 7—figure supplement 3**), in agreement with the fact that the identically tagged genomic construct rescues the null mutant to full viability.

## Asymmetric EGFR signaling is essential for axon branch pruning through regulating actin localization

To analyze downstream players of EGFR-dependent refinement we first focused on the canonical EGFR pathway. Activation of the MAPK cascade and transcriptional changes in the nucleus are main features of this pathway (**Vivekanand and Rebay, 2006**). A nuclear marker for active MAPK signaling is double phosphorylated ERK (dpERK). Despite the fact that we verify expression in developing L3 eye disc (**Figure 8—figure supplement 1A**), we were not able to detect dpERK in developing DCN (**Figure 8—figure supplement 1B-B″**). One caveat is that activation of ERK might be difficult to detect due to low expression levels and timing issues. To further investigate if the canonical pathway is involved, we analyzed the effect of MAPK pathway genes (**Vivekanand and Rebay, 2006**) on DCN axon refinement. Expression of Ras1$^{RNAi}$, Drk $^{RNAi}$, a constitutively active

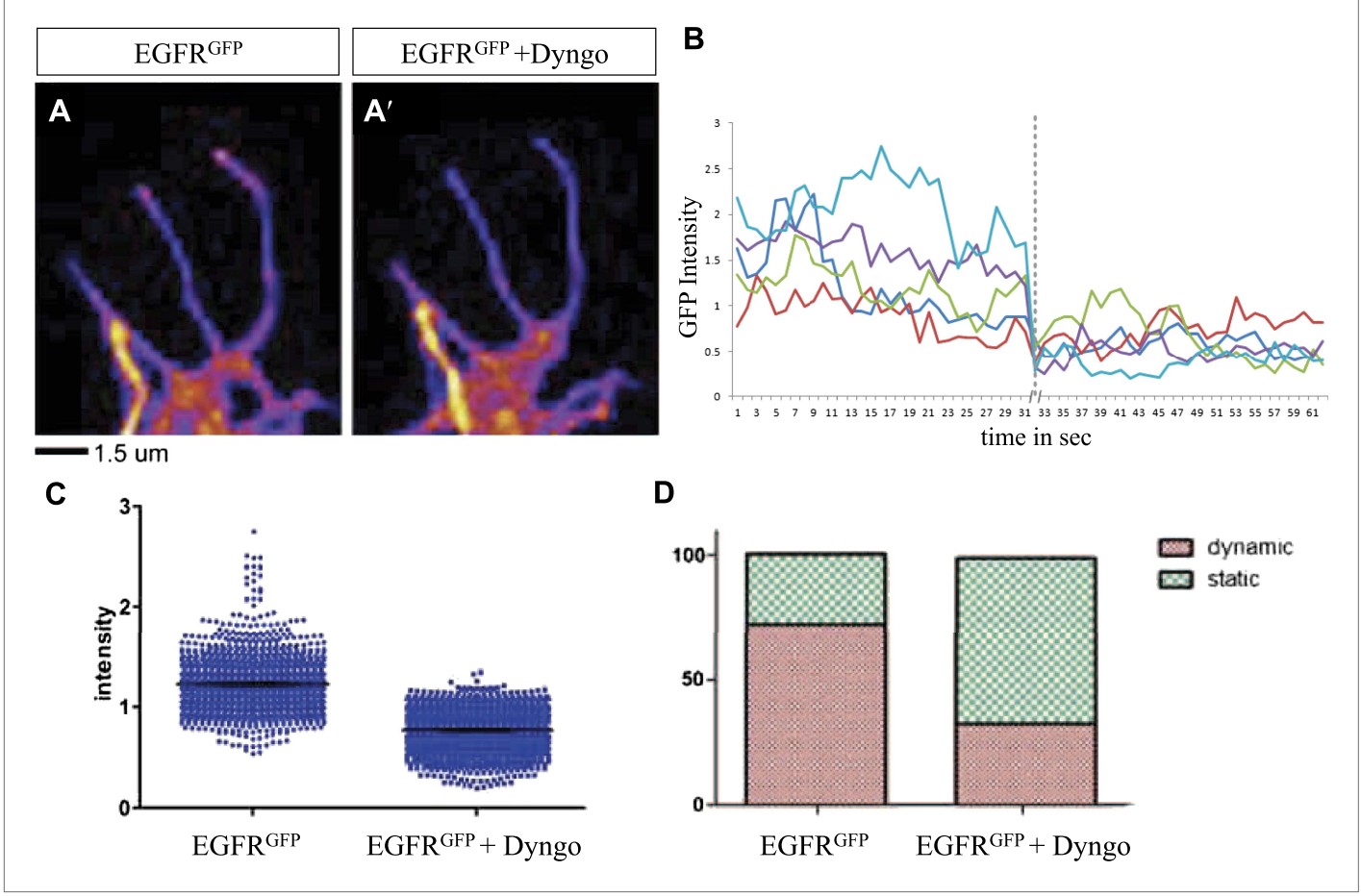

**Figure 6**. Differential EGFR localization in filopodial of primary Drosophila neurons requires endocytosis. (**A–A'**) UAS-EGFR[GFP] expressed with elav-Gal4 driver in primary Drosophila neurons. False color image displaying a heat map of an EGFR[GFP]-expressing growth cone before (**A**) and after (**A'**) treatment with Dyngo. (**B**) Maximal intensity of EGFR[GFP] in filopodia within one neuron over time (2 min), before and after treatment (indicated by dotted line) with Dyngo. (**C**) Scatter plot from EGFR[GFP] maximal intensities from filopodia from 6 neurons, showing a significant decrease in levels after treatment with Dyngo (EGFR[GFP] maximal intensities in DMSO: 1.229 ± 0.0074, n = 1403; EGFR[GFP] maximal intensities in Dyngo: 0.768 ± 0.005, n = 1401, p<0.001). (**D**) Effect of Dyngo on filopodia dynamics. Quantification of static vs dynamic (extensions and retractions) behaviors of filopodia shows a significant distribution change between controls (EGFR[GFP]-expressing neurons in DMSO) and Dyngo-treated EGFR[GFP]-expressing neurons (control: static = 28, dynamic = 72; Dyngo treated: static = 66, dynamic = 32, p<0.001). Mann–Whitney test. ***p<0.001.

form of ERK or a constitutively active form of Ras1 did not change the DCN branching pattern (***Figure 8—figure supplement 1C–G***). These results indicate that refinement occurs independently of the canonical EGFR pathway.

The fast growth and retraction rates of axonal branches in wildtype brains, altered growth dynamics upon EGFR inhibition and the well-established role for cytoskeletal proteins in branch formation (***Gallo, 2011***) together suggest that EGFR activation may act via cytoskeleton regulation in this case. We used actin-GFP (***Verkhusha et al., 1999***) and Utrophin-GFP (***Rauzi and Lenne, 2011***) expression in the DCNs to examine the distribution of total actin and filamentous actin (F-actin), respectively, in wild type vs EGFR[DN] backgrounds. The F-actin binding protein Utrophin (***Galkin et al., 2002***) was utilized to analyze the distribution of actin filaments in wildtype and EGFR[DN] axons. Utrophin-GFP reveals that F-actin is largely confined to the branches (***Figure 8A–A'''***, arrowheads) with low levels of F-actin in the axon shafts (***Figure 8A'''***, arrow) of wildtype brains. In contrast, in DCNs expressing EGFR[DN] F-actin distribution appears weaker and more diffused over the axon shaft and axon branches (***Figure 8B–B'''***, arrowheads). Similar to F-actin, total actin-GFP concentrates at the branch tips (***Figure 8—figure supplement 2A–A'''***, arrowheads) and little actin is present within

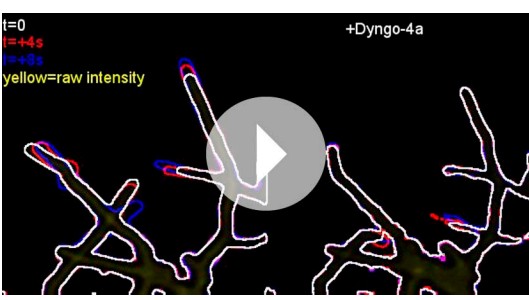

t=0
t=+4s
t=+8s
yellow=raw intensity

+Dyngo-4a

**Video 4**. Comparison egfr vs egfr + dingo. This video is related to **Figure 6**. Video shows the side-by-side comparison of dynamic behavior of filopodia with and without Dyngo-4a (a dynamin inhibitor) treatment. The intensity of the EGFR-GFP signal is displayed in yellow. The outlines of the filopodia have been segmented by subsequently thresholding and outline detection of the fluorescent signal. To show the dynamic behavior of the filopodia in the time-lapse video, the outlines of the following two frames (4 and 8 s ahead of the current frames) are displayed in red and in blue respectively. The untreated filopodia move more than the Dyngo-4a treated ones.

the main axon shaft. In contrast, in EGFR[DN] DCNs total actin also accumulates in blebs along the entire length of the axons and their branches (**Figure 8—figure supplement 2B–B'''**, arrows). Axonal swellings have considerably lower F-actin accumulation (**Figure 8B'''**, asterisk) compared to total actin (**Figure 8—figure supplement 2B'''**, asterisk) suggesting that in EGFR[DN] axons, monomeric actin is retained in axonal swellings along the axons, thus potentially inhibiting efficient actin polymerization dynamics at the branch tips.

Our data thus far suggest a model whereby dynamic localization of the EGFR results in differential signaling between developing filopodia and axonal branches. This enhances actin dynamics and results in the proper balance of branch growth and pruning. However, an alternative possibility is that EGFR signaling simply instructs branch retraction. Both models predict increased branch numbers when EGFR signaling is inhibited. However, if EGFR signaling instructs branch pruning, activated EGFR would result in reduced axonal branching. In contrast, if EGFR signaling asymmetry is indeed required for the correct number of DCN axonal branches, then constitutive activation of the EGFR should also result in increased axonal branching. To distinguish between the two models, we analyzed the effect of a constitutively active form (UAS-EGFR[CA]). In agreement with a differential local signaling model, EGFR[CA] induces a significant increase of DCN branches both in vitro (**Figure 8—figure supplement 3**) and in vivo (**Figure 8C–D**) similar to down regulation of EGFR signaling. Importantly, similar to loss of EGFR function, the increase in branch number induced by gain of EGFR function is also due to reduced pruning during development (**Figure 8E–H**). Furthermore, in EGFR[CA] axons total actin and F-actin also distribute more uniformly across the axonal projection (**Figure 8I–I'''**, **Figure 8—figure supplement 2C–C'''**, arrows), again suggesting reduction of efficient polymerization dynamics. In summary, EGFR signaling affects branch growth and retraction likely through the regulation of actin polymerization.

## Discussion

The refinement of exuberant branches is a crucial step during the development of a neuronal network. In this work, we exploit an adult-specific model circuit, the dorsal cluster neurons, to study developmental neurite pruning processes in the CNS of *Drosophila*. DCN axons form a stereotyped number of branches innervating the medulla through initial excessive axon branch formation followed by a refinement process. Our data suggest a model (**Figure 9**) whereby uneven distribution of EGFR to developing DCN axonal branches is required to eliminate exuberant branches and help generate the correct adult connectivity pattern.

During mammalian development neurites are generally formed in excessive numbers and subsequently refined to form the mature circuit (**Low and Cheng, 2006**). This mechanism ensures that all targets are properly innervated, it enables further specification of connections by the target environment like neighboring neurons and glia (**Stevens et al., 2007**) and permits the removal of exuberant or mistargeted branches. Studying real-time events in the mammalian system involving CNS refinement is challenging. The *Drosophila* developing brain culture system used in this work combined with live imaging allows examination and manipulation of neuronal growth dynamics. Our data suggest that EGFR signaling, in part triggered by the co-innervation of the target neuropil by sensory neurons from the retina, is a crucial determinant of axonal branch refinement by the regulation of filopodial growth and retraction dynamics. Finally, we find that EGFR activity regulates actin polymerization dynamics at the branch tips. Consistent with this notion, we find that interfering with actin dynamics in vivo by

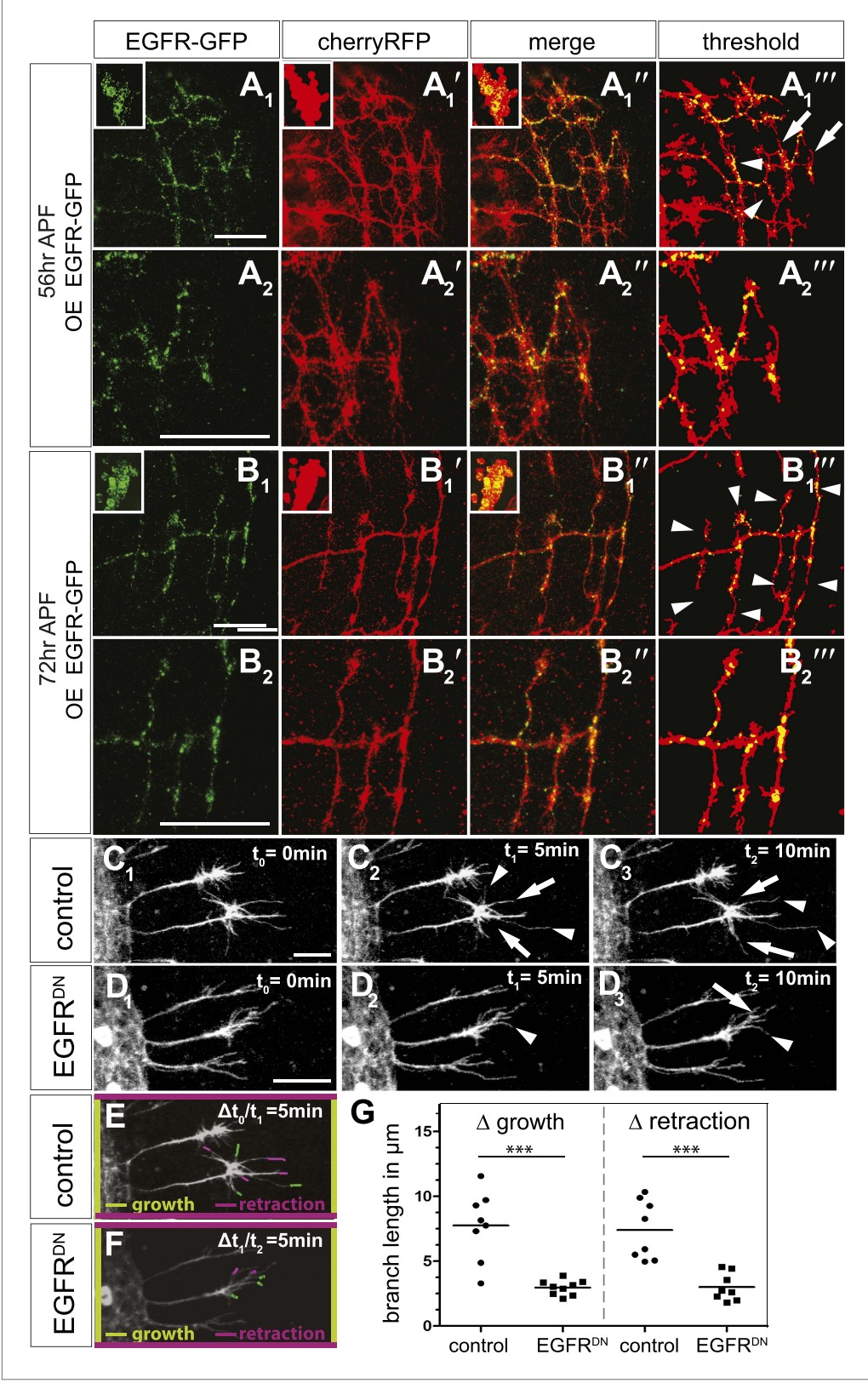

**Figure 7**. EGFR mediates a probabilistic branch refinement process. (**A–B**) EGFR localization examined by expressing UAS-EGFR[GFP] (green) in the DCNs (red, UAS-cherryRFP) during pupal development at (**A**) 56 hr APF

*Figure 7. Continued on next page*

*Figure 7. Continued*

and (**B**) 72 hr APF. EGFR$^{GFP}$ expression was observed in a punctate pattern in the cell bodies (insets in **A** and **B**) and along the axonal branches (**A** and **B**). Images **A/B** and **A'/B'** were subjected to thresholding and merged (**A'''/B'''**). Differential localization results in branches with (**A'''**, arrowheads) and without (**A'''**, arrows) EGFR$^{GFP}$ at 56 hr APF, whereas most if not all branches contain EGFR$^{GFP}$ at 72 hr APF (**B'''**, arrowheads). High magnification shows EGFR localization at branches at 56 hr APF (**A$_2$**) and 72 hr APF (**B$_2$**). (**C**) Z-stack projections from live imaging time-lapse videos of control axons at around 40 hr APF between $t_0$ = 0 min (**C$_1$**) and $t_2$ = 10 min (**C$_3$**) with 5-min intervals. (**D**) Z-stack projections from live imaging time-lapse videos of EGFR$^{DN}$ axons at around 40 hr APF between $t_0$ = 0 min (**D$_1$**) and $t_2$ = 10 min (**D$_3$**) with 5 min intervals. Arrows indicate branches being pruned while arrowheads point to growing branches. (**E**) Visualization of growth (green) and retraction (purple) events between $t_0$ = 0 min (**C$_1$**) and $t_1$ = 5 min (**C$_2$**) in control. (**F**) Visualization of growth (green) and retraction (purple) events between $t_1$ = 5 min (**D$_2$**) and $t_2$ = 10 min (**D$_3$**) in EGFR$^{DN}$. (**G**) Quantification of growth and retraction dynamics at branches using the tracer tool shows significant decrease in branch lengths in EGFR$^{DN}$ compared to control. Control (growth) 7.75 ± 2.65 (n = 8), EGFR$^{DN}$ (growth) 2.97 ± 0.56 (n = 9, p<0.001). Control (retraction) 7.4 ± 2.28 (n = 8), EGFR$^{DN}$ (retraction) 3 ± 1.08 (n = 8, p<0.001). Horizontal lines represent the mean for each data set. *t* test. ***p<0.001. The scale bars represent 20 μm.

The following figure supplements are available for figure 7:

**Figure supplement 1**. Localization of EGFR.

**Figure supplement 2**. DCN branch pattern in cultured pupal brains.

**Figure supplement 3**. UAS-EGFR$^{GFP}$ localizes and functions similar to endogenous EGFR.

**Figure supplement 4**. Overexpression of wild-type EGFR does not cause a significant increase in axonal branching.

inhibition of the small GTPase RhoA or constitutive activation of the actin filament severing protein Cofilin, is sufficient to cause ectopic axon branch formation in the DCNs (data not shown). EGFR expression has been observed in neurites of mammalian neurons (*Gerecke et al., 2004*; *Chen et al., 2005*) and knock-out of the EGFR in the mouse results in increased neurite branching in the skin (*Maklad et al., 2009*), suggesting that the mechanism we identify in the fly CNS may be more generally utilized.

In summary, we report evidence for the notion that differential branch signaling is a determinant of connection specificity. We show that intrinsically asymmetric EGFR localization and signaling is required for efficient branch pruning. Several lines of evidence support this conclusion. First, EGFR is asymmetrically localized in branches and filopodia both in vivo and in cultured primary neurons. Second, both inhibition and constitutive activation result in failure of axonal branch refinement. Third, overexpression of the wildtype receptor, which is differentially localized and trafficked, is not sufficient to produce a phenotype. This argues that receptor localization dynamics—possibly mediated by endocytosis—rather than total EGFR levels, is the cue for filopodial collapse and subsequent axonal branch pruning. What explains the link between regulation of dynamic behavior and the generation of a specific number of axonal branches? A hint to this comes from three observations. First, both loss and gain of EGFR function increase proportion of static filopodia from less than 10% to more than 30%, subsequently increasing the number of axonal branches. Second, this filopodial behavior correlates with small, but significant and highly dynamic differences in EGFR localization. Third, loss of EGFR signaling increases the variability in axon branch number. Based on these observations we propose that in wildtype neurons most dynamic filopodia collapse over time, resulting in continuous redistribution of EGFR among fewer and fewer remaining filopodia. This process stops usually when only one filopodium remains at a given branching point, and occasionally when EGFR happens to distribute equally between the last two filopodia. This probabilistic process does not require an additional mechanism of branch 'tagging and selection' and can explain both EGFR loss of function phenotypes: increased branch number and increased variability. What remains to be determined is the interaction between EGFR-dependent branch dynamics and the specificity of the spatial pattern of branches.

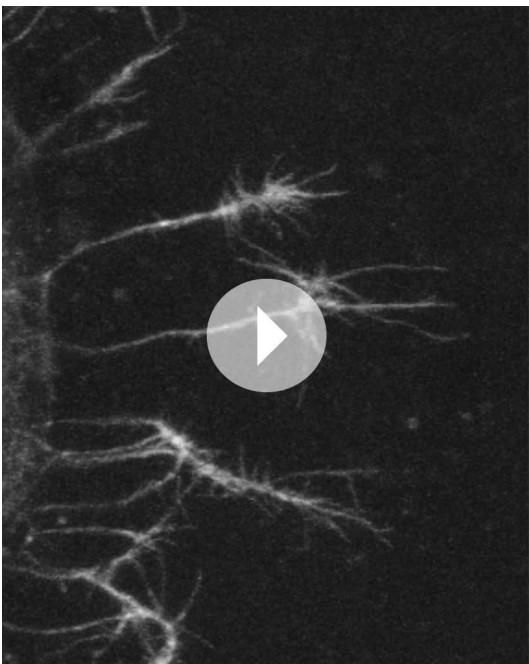

**Video 5**. brain culture WT 40 hr. This video is related to *Figure 7*. Live imaging time-lapse videos of control axons at around 40hr APF. Corresponds to images presented in *Figure 7C* and quantified in *Figure 7G*. Images were collected every 5 min for 45 min.

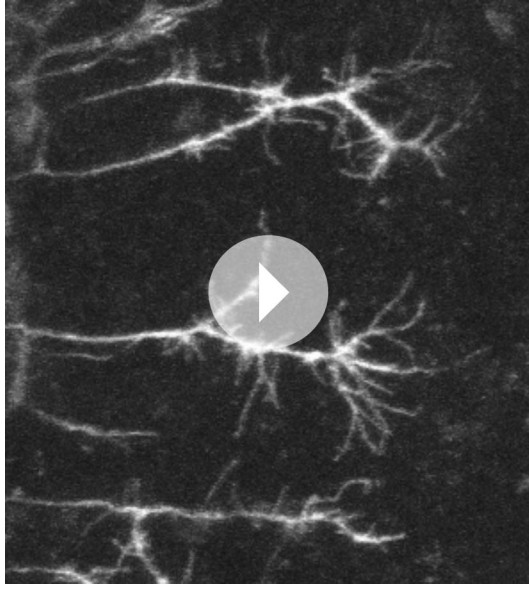

**Video 6**. brain culture EGFR-DN 40 hr. This video is related to *Figure 7*. Live imaging time-lapse videos of EGFR[DN] axons at around 40hr APF. Corresponds to images presented in *Figure 7D* and quantified in *Figure 7G*. Images were collected every 5 min for 40 min.

## Materials and methods

### Fly strains and genetic manipulation

Fly stocks were cultured on standard fly food. All experiments were performed in temperature-controlled incubators at 25°C or 28°C. The GAL4 driver lines used in this study are: ato-Gal4-14a (*Hassan et al., 2000*), sca-Gal4, elav-Gal4. The UAS-reporter stocks were the following: UAS-CD8-GFP, UAS-CD8-cherryRFP, UAS-LacZ, UAS-EGFR[DN-A] (gift from M Freeman), UAS-EGFR[DN-B], UAS-EGFR[RNAi] (VDRC107130), UAS-Spi[RNAi] (TRiP, JF03322), UAS-EGFR[CA], UAS-Utrophin-GFP (gift from T Lecuit), UAS-Moesin-GFP (*Dutta et al., 2002*), lexAop-myr-GFP, ato[lexA]. Additional fly stocks and mutants used were: Canton-S, EGFR[T1]. For FLP-out system experiments *yw, hs-FLP; UAS-FRT CD2, y FRT mCD8::GFP; atoGal4-14a, UAS-LacZ* was crossed out to *Canton-S* or UAS-EGFR[DN-A]. ato[lexA] was created by knocking LexA into the *ato* locus to drive LexAop-myr-GFP expression.

### *Drosophila* primary neuron cultures

*Drosophila* primary neuron cultures were generated as described previously (*Sanchez-Soriano et al., 2010*; *Prokop et al., 2011*). In brief, stage 11 embryos (6–7 hr AEL at 25°C) were homogenised, treated for 5 min at 37°C with dispersion medium, washed and dissolved in Schneider's medium. Then, the aliquots were transferred to coverslips, kept as hanging drop cultures in air-tight special culture chambers (*Deak et al., 1980*) for 6 hr or 4 days at 26°C. Live imaging of primary neurons was performed on a Delta Vision (RT) (Applied Precision, Issaquah, WA) restoration microscope using a (100 × 3 phase) objective and the (Sedat) filter set (Chroma Technology, Germany). The images were collected using a Coolsnap HQ (Photometrics, Tuscon, AZ) camera, image acquisition was through Softworx. For immunocytochemistry, cells were fixed (30′ in 4% paraformaldehyde in 0.05 M phosphate buffer, pH 7.2), washed in PBS 0.1% Triton X-100 (PBT), then incubated with antisera diluted in PBT.

### Inhibition of endocytosis and quantification

To inhibit endocytosis, cells were incubated for 6 min with 0.14 mM dynamin inhibitor Dyngo-4a (Abcam), diluted in Schneider's medium from stock solution in DMSO. For controls, equivalent concentrations of DMSO were diluted in Schneider's medium. The effect of the dynamin

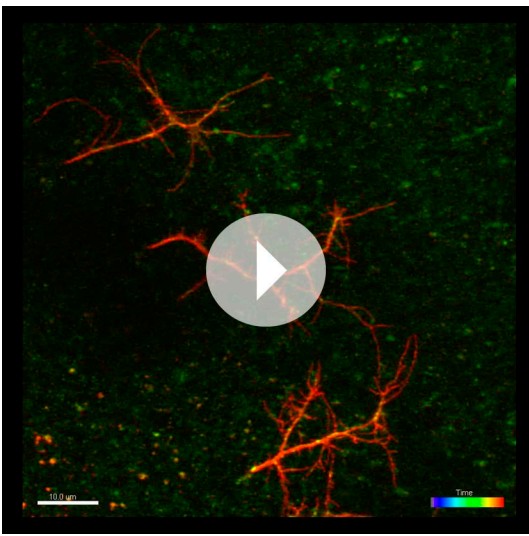

**Video 7**. brain culture EGFR-GFP. This video is related to **Figure 7**. EGFR shows differential and dynamic localization in developing dorsal cluster neurons in vivo. UAS-CD8-RFP and UAS-EGFR-GFP were expressed with ato-Gal4 in wildtype *Drosophila* brains. Intact eye–brain complexes were imaged live at 45% APF. Maximum projection images demonstrating three DCN axon terminals in a time-lapse video of 3 hr with 2 min time intervals. Extension of the axons over time could be observed especially in the upper axon. All axons demonstrate rapid filopodial dynamics as well as changes in EGFR localization over time.

inhibitor Dyngo-4a on levels of EGFR$^{GFP}$ was quantified in FIJI, by measuring the maximal intensities at the distal ends of filopodia during 2 min before and after drug treatment. Previous to quantification, the background of acquired images was subtracted (atrous wavelet transform, scales 1–8 minus low pass image). The GFP intensity of each filopodia was normalized to the mean of maximal intensities of all filopodia within a cell before and after treatment.

## Immunohistochemistry

The following primary antibodies were used in the in vivo experiments: rabbit anti-GFP (1:1000; Invitrogen), mouse anti-GFP (1:500; Invitrogen), mouse anti β-galactosidase (1:1000; Promega), rabbit anti β-galactosidase (1:1000; Cappel), mouse MAb 24B10 anti-Chaoptin (1:200; DSHB), rabbit anti-DsRed (1:500; Clontech), mouse anti-NC82 (1:100; DSHB), rat anti-DN cadherin (1:20; DHSB DN-EX#8). The following primary antibodies were used in the in vitro experiments: mouse anti-tubulin (1:1000; Sigma), goat anti-GFP (1:1000; Abcam). The incubation with the primary antibodies was followed by several washes in PBT (1 hr) and a final incubation with the appropriate fluorescent secondary antibodies (in vivo: *Alexa* 488, 555 or 647, Molecular Probes, 1:500, in vitro: FITC- or Cy3-conjugated affinity-purified secondary antibodies (donkey, 1:200 [Jackson ImmunoResearch])).

In vitro filamentous actin was detected with TRITC-conjugated phalloidin (Sigma). After several washes in PBT the samples were mounted in vectashield.

## Imaging

Confocal stacks of fixed brains were made using Zeiss LSM 510 or Leica SP6 confocal microscopes. Neuronal cell culture imaging was conducted with an AxioCam camera mounted on an Olympus BX50WI microscope. DCN live imaging was conducted with a Leica SP6 resonance scanning confocal microsocope. In general, a confocal stack comprising the axonal projection of Dorsal Cluster Neurons (30–40 single projections) was recorded every 5 min. Resonance scanning allowed high scan speed with lower laser intensities and therefore ensures preservation of living tissue due to decreased phototoxicity. Projection images were generated and further processed with ImageJ. For tracking of axon branches we have used the 'simple neurite tracer' a plugin for ImageJ from Mark Longair (Fiji, http://pacific.mpi-cbg.de).

## Quantification of developmental branches

Images of medulla axons were skeletonized and subsequently automatically analyzed using the 'Skeletonize3D' and 'AnalyzeSkeleton' free plugins for ImageJ/FIJI (freely downloadable from the FIJI website: URL: http://pacific.mpi-cbg.de/wiki/index.php/Fiji). Number of developmental branches is the number of end points from the skeleton.

## Whole pupal brain culture system and live imaging

Staged pupal brains were dissected in cold Schneider's Drosophila Medium (GIBCO) and transferred to the culture plate inserts and cultured according to the whole brain explant system described previously (*Ayaz et al., 2008*). After allowing the pupal brains to attach to the membrane of the culture plate insert for a minimum of 8 hr, the membrane was cut out of the plastic insert and carefully transferred to a closed confocal imaging perfusion chamber (Harvard IC30 confocal

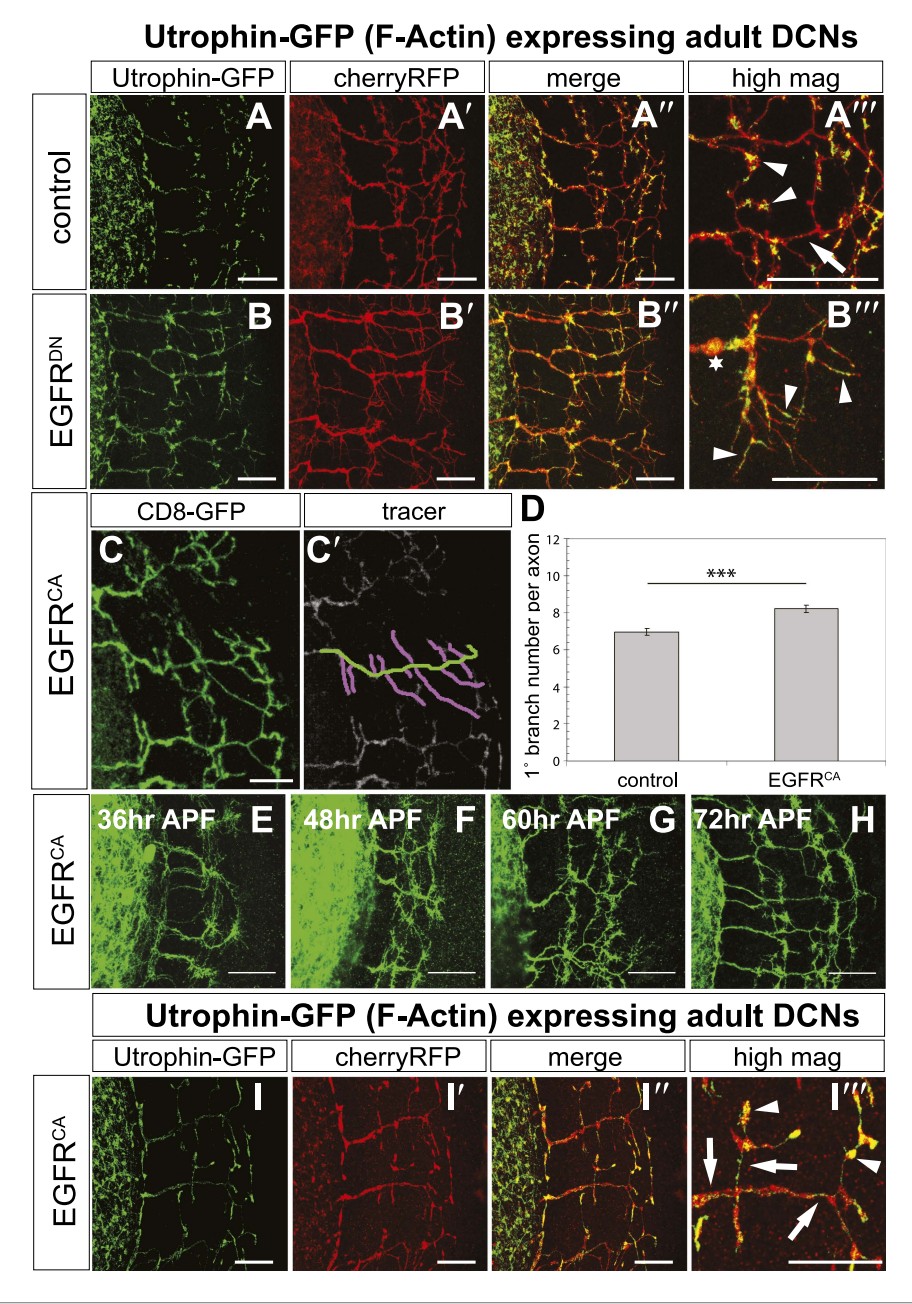

**Figure 8**. EGFR regulates actin polymerization in DCN axonal branches. (**A–B**) Utrophin (F-actin) localization in adult DCN (red, UAS-cherryRFP) by expressing UAS-utrophin-GFP (green) in (**A**) control and (**B**) EGFR[DN]. (**A′′′–B′′′**) High magnification of the branch tips. Arrowheads show localization of Utrophin at the branch tips. Arrows show Utrophin localization along axon shafts. Asterisk in (**B′′′**) shows weak Utrophin accumulation in axonal swellings. (**C**) Overexpression of a constitutively active form of EGFR (UAS-EGFR[CA]) results in increased branching in the adult DCN. (**C′**) Visualization of branches (purple) along a single main axon shaft (green), using the tracing tool. (**D**) Quantification of adult primary branch numbers per axons shows significant increase of branches in EGFR[CA] compared to control. Control 6.96 ± 1.34 (n = 60), EGFR[CA] 8.22 ± 1.47 (n = 55, p<0.001). Error bars represent SEM. Mann–Whitney test. ***p<0.001. (**E–H**) Axonal branch pattern at different pupal stages shows excessive branching during mid-pupal development. Branch morphology at (**E**) 36 hr APF, (**F**) 48 hr APF, (**G**) 60 hr APF, and (**H**) 72 hr APF. (**I–I′**) Utrophin (F-actin) localization in adult DCN (red, UAS-cherryRFP, **I′**) by expressing UAS-utrophin-GFP (green, **I**) in an EGFR[CA] background. (**I′′**) Merge of DCNs (red) and Utrophin (green). (**I′′′**) High magnification of the branch tips.
*Figure 8. Continued on next page*

*Figure 8. Continued*

Arrowheads show localization of Utrophin at the branch tips. Arrowheads show localization of Utrophin at the branch tips. Arrows show Utrophin localization along axon shafts. The scale bars represent 20 μm.
The following figure supplements are available for figure 8:

**Figure supplement 1**. The canonical MAPK pathway is not involved in DCN refinement.

**Figure supplement 2**. EGFR regulates actin polymerization in DCN axonal branches.

**Figure supplement 3**. Branch increase in cultured neurons by expression of EGFR[CA].

imaging chamber) connected to a peristaltic pump that slowly perfuses culture solution over the live tissue. A fast resonant scanning confocal microscope (Leica TCS SP5) with special high-aperture immersion lenses was used to allow three-dimensional recordings over time at faster frame rates which reduces phototoxicity. Live imaging was performed as previously described (*Williamson and Hiesinger, 2010*).

## Quantification of live imaging

For tracking of growth and retraction dyamics we have used the 'simple neurite tracer' a plugin for ImageJ from Mark Longair (Fiji, http://pacific.mpi-cbg.de). We have traced dynamic axon branches by using the tip of an axon at time point $t_0$ as starting point and the tip of the same axon at time point $t_1$. The length of the resulting fragment represents the length of the growing or retracting axon.

## Generation of UAS-EGFR[GFP] transgenic flies

UAS-EGFR[GFP] was created by fusing the *Drosophila egfr* cDNA from the pUC13-DERII construct (*Schejter et al., 1986*) and *eGFP* cDNA (Clontech) from pStinger into *pUAST-Attb* vector (Genbank EF362409.1). Two Gly-Gly-Ser bridges (GGSGGS) have been introduced between the two open reading frames. Transgenic flies were created at GenetiVision Inc. (Houston, USA) using PhiC31-mediated transgenesis in the VK37 docking site (2L, 22A3) and in the VK31 docking site (3L, 62E1).

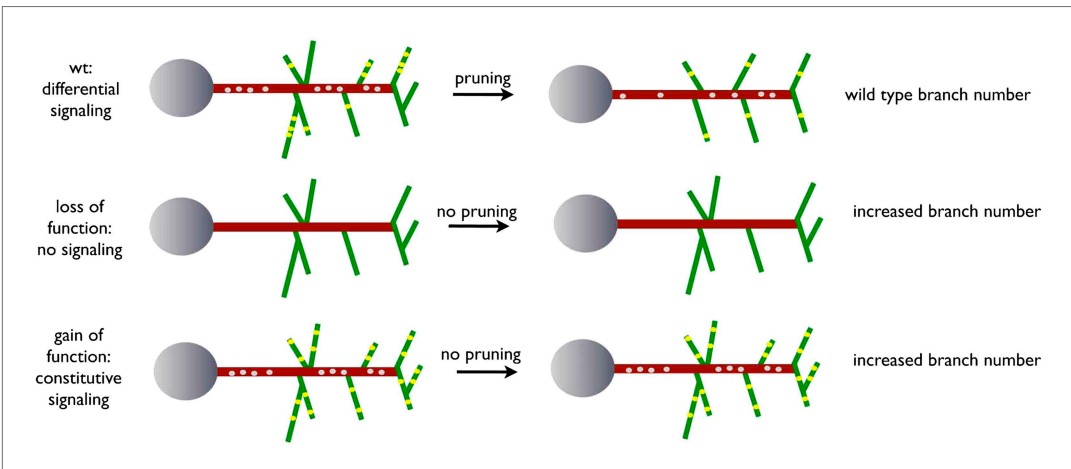

**Figure 9**. A model for EGFR function in axonal branching. Local asymmetries in tyrosine kinase receptor activity in axonal branch, driven by differential distribution of active receptor molecules in filopodia, generate dynamical behavior and drive branch pruning. Gray dots represent EGFR puncta trafficked along the axon shaft (red) while yellow dots represent active EGFR puncta within branches (green).

## EGFR<sup>GFP</sup> measurements in cultured neurons

Levels of *EGFR$^{GFP}$* in static vs dynamic filopodia were quantified in FIJI, by drawing a box at the distal ends of filopodia and measuring the maximal intensities within. Only neurons with both static and dynamic filopodia were used for the analysis, and the GFP intensity of each filopodia was normalized to the mean of maximal intensities of all filopodia within a cell. For dynamic filopodia, measurements were taken during the first 8 s of the retraction or extension. For static filopodia, measurements were taken during 8 s at the middle of the recording period. These measurements were used to calculate the ratio of EGFR$^{GFP}$ in dynamic minus static filopodia (GFP maximal intensity of each dynamic phase minus the mean of GFP maximal intensity in static filopodia).

## Statistical tests

For non-normally distributed samples the nonparametric ANOVA Kruskal–Wallis test with Dunn's multiple comparisons for *Figure 2E* and the Mann–Whitney test for *Figure 8D* was performed. Student's *t* test was used for *Figures 3P and 7G*. For neuronal culture experiments, the Mann–Whitney test was used for *Figure 4C,F, 5C and 6D*.

## Materials and methods for figure supplements and videos

### Fly strains and genetic manipulation

The additional GAL4 driver line was: GMR-Gal4 and Dpp-Gal4. The UAS-reporter stocks were the following: UAS-nSyb-GFP, UAS-Syt-GFP, UAS-ERK$^{CA}$, UAS-Ras1$^{CA}$, UAS-Ras1$^{RNAi}$, UAS-Drk$^{RNAi}$, UAS-Actin-GFP, UAS-EGFR. Additional fly stocks and mutants used were: Vein-lacZ (gift from I Miguel-Aliaga), EGFR$^{1k35}$, Spi$^{scp2}$.

For MARCM experiments (*Lee and Luo, 1999*) *yw; hsflp, UAS-CD8-GFP; FRT42D Tub-Gal80/ CyO; atoGal4-14a/TM6c* were used in conjunction with *yw; FRT42D EGFR$^{1k35}$/ CyO*. The crosses were set up at 25°C and transferred every day. 2 to 4 days after egg laying the samples were heatshocked for 3 hr at 37°C and shifted back to 25°C until eclosion.

### Immunohistochemistry

The following primary antibodies were used in this study: rat anti-DN cadherin (1:20; DHSB DN-EX#8), rat anti-EGFR (1:1,000, from B Shilo), rabbit anti-dpERK (1:100; Cell Signaling), mouse anti-NC82 (1:100; DSHB), Rabbit anti Rab5 (1:500; Abcam), Rat anti Rab11 (*Dollar et al., 2002*, 1:500).

### *Drosophila* long-term pupal brain culture

Culture medium was modified from *Ayaz et al. (2008)*. The culture medium contained 5000 U/ml penicillin, 5 mg/ml streptomycin, 10% fetal bovine serum, 20 µg/ml insulin and 2 µg/ml of ecdysone in Schneider's Insect Medium. Pupal brains were dissected in room temperature culture medium and immediately placed in a sterile culture dish containing fresh medium. Brains remained undisturbed in the dark at 25°C throughout the culture period. At the end of the culture period the brains were rinsed briefly in PBS and then fixed in 2% paraformaldehyde for 1 hr followed by standard Immunohistochemistry.

### Live culture and imaging of EGFR<sup>GFP</sup> in DCNs

Intact pupal eye–brain complexes dissected from 45% APF *Drosophila* were cultured in a Schneider's based medium (*Ayaz et al. 2008*), immobilized in 0.4% agarose solution. Confocal stacks of DCN terminals were captured every 2 min using a Leica SP5 resonant scanner for 3 hr, with a 63X (NA = 1.3) glycerol objective. Images were deconvolved using Autoquant X3 (Media Cybernetics) and analyzed with Imaris 7.6 (Bitplane).

## Acknowledgements

We thank the Bloomington Stock Center, the Transgenic RNAi Project at Harvard Medical School (TRiP) and Vienna *Drosophila* RNAi Center (VDRC) for providing *Drosophila* Stocks. We thank B Shilo for sharing the EGFR antibody and the pUC13-DERII construct, and Gerald M Rubin for support and discussions on brain culture methodology. Additionally we thank J Kasprowicz and S Kuenen, and members of the BAH lab for technical assistance and constructive comments, Egor Zindy for help with image analysis, Andreas Prokop and Patrick Caswell for helpful advice and Marian Wilkin and Sean Sweeney for reagents. We thank Dietmar Schmucker for stimulating discussions and comments on the manuscript.

## Additional information

### Funding

| Funder | Grant reference number | Author |
|---|---|---|
| Vlaams Instituut voor Biotechnologie | | Marlen Zschätzsch, Carlos Oliva, Marion Langen, Natalie De Geest, Alessia Soldano, Sebastian Munck |
| Fonds Wetenschappelijk Onderzoek | G.0543.08, G.0680.10, G.0681.10, G.0503.12 | Marlen Zschätzsch, Carlos Oliva, Marion Langen, Natalie De Geest, Alessia Soldano, Sebastian Munck |
| Belgian Federal Science Policy Office | | Marlen Zschätzsch, Carlos Oliva, Marion Langen, Natalie De Geest, Alessia Soldano, Sebastian Munck |
| KU Leuven | | Marlen Zschätzsch, Carlos Oliva, Marion Langen, Natalie De Geest, Alessia Soldano, Sebastian Munck |
| Wellcome Trust | 087742/Z/08/Z | Natalia Sanchez-Soriano |
| Biotechnology and Biological Sciences Research Council | BB/I002448/1 | Natalia Sanchez-Soriano |
| Howard Hughes Medical Institute | | William C Lemon |
| University Of Manchester | | Natalia Sanchez-Soriano |
| CONICYT postdoctoral fellowship | | Carlos Oliva |
| National Institutes of Health | RO1EY018884 | P Robin Hiesinger |
| Welch Foundation | I-1657 | P Robin Hiesinger |

The funders had no role in study design, data collection and interpretation, or the decision to submit the work for publication.

### Author contributions

MZ, NS-S, Conception and design, Acquisition of data, Analysis and interpretation of data, Drafting or revising the article; CO, NDG, WRW, AS, Acquisition of data; ML, MNÖ, WCL, Acquisition of data, Analysis and interpretation of data; SM, Analysis and interpretation of data; PRH, Analysis and interpretation of data, Drafting or revising the article; BAH, Conception and design, Analysis and interpretation of data, Drafting or revising the article

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
