## [Decision Letter]

Thank you for sending your work entitled “Regulation of branching dynamics by axon-intrinsic asymmetries in Tyrosine Kinase Receptor signaling” for consideration at *eLife*. Your article has been favorably evaluated by a Senior editor and 2 reviewers, one of whom is a member of our Board of Reviewing Editors.

The Reviewing editor and the other reviewer discussed their comments before we reached this decision, and the Reviewing editor has assembled the following comments to help you prepare a revised submission.

In the present study, Zchatzch and collaborators demonstrate a novel role for EGF receptor signaling in axon branching. Using Drosophila adult brain neurons as a model, the authors provide comprehensive evidence for the instrinsic role of EGFR in regulating terminal axon branching by controlling the dynamics of filopodia. This evidence is based on a comprehensive set of experimental approaches including genetic gain- and loss-of-function, single cell analysis and time-lapse microscopy using an innovative ex vivo preparation. Furthermore, the authors explore the downstream signaling pathway that might underlie this unconventional function for EGFR and demonstrate that it is not involving a 'canonical' set of tyrosine kinase receptor signaling (Ras-MAPK signaling). Finally, the authors explore the involvement of actin dynamics and show that EGFR signaling in axon branching involves a control of F-actin polymerization in axonal filopodia.

This article is clearly providing novel insights into the cellular and molecular mechanisms underlying axon branching in vivo, which is a highly significant aspect of brain wiring. Therefore, the significance of results is high. However, one reviewer raised a number of substantial concerns regarding some aspects of the work, focusing on: First, the loss and gain of function of EGFR causes the same effect in branching. It is true that there might be a way to interpret this, but the understanding of the downstream signaling events is not sufficient to validate the model. Second, endocytosis blockade led to a decrease of the receptor level. This is not intuitive and needs to be explored further. Third, the subcellular localization of EGFR is very interesting. However, the correlation between the presence of EGFR in vivo and the fate of branches is not established.

Therefore, we would like to have the authors respond to the major comments listed below. If they can provide experimental evidence for most of the major points (with a strong priority for point 4 below), we will be happy to consider a revised version.

Major comments:

1) Videos 1 and 2 seem to show very different modes of movements. In Video 1, it seems that vesicular transport can be seen. Video 2 shows a much more diffused staining pattern. The intensity changes in this video seem to be fluctuations instead of discrete vesicular movements.

2) It is usually assumed that inhibition of endocytosis would lead to an increase in the level of receptors on the membrane. Figure 6 showed the opposite. First, is it known whether the EGFR is mostly on the surface or internal? The main limitation of using a simple EGFP fusion protein is that the authors cannot distinguish between surface receptor from internal stores present in endosomes/exosomes. Second, it is entirely possible that the authors are correct and some type of transcytosis (which requires endocytosis) is sending EGFR to the filopodia. Is there any redistribution of EGFR from the cell body or primary axon to the branches?

3) In Figure 7, the authors use overexpressed EGFR-GFP to study the subcellular localization of EGFR. Does this construct cause there any effect on the branching phenotype? In the EGFR^DN^ background, does the EGFR-GFP rescue the phenotype? This is of course trying to test if the EGFR-GFP construct is functional.

4) The critical experiment is to understand the correlation of having strong EGFR-GFP puncta on a branch and whether they will be pruned in Figure 7. We did not get a clear answer whether that has been attempted or not from the text.

5) In Figure 7, both growth and retraction of neurites are severely affected by the DN construct. The effects on the growth of the neurite are inconsistent with the lack of outgrowth phenotype in the loss-of-function analyses, which challenges the relevance of this long-term culture system.

6) It is puzzling that the gain-of-function EGFR causes the same phenotype as the loss-of-function. Although the authors have a way of thinking about this result, detailed analyses of the EGFR^CA^ during the developmental course is necessary to test if the loss- and gain-of-function manipulation indeed has the same effects in growth and pruning.

7) It is unclear what is the cellular source of EGF ligand(s). The results implicate that EGF ligand(s) could act in an 'axon-autonomous' way to control branching through filopodia through contacts between filopodia of adjacent axons. However, the authors never document where EGF ligand(s) are located. Are there expressed by the axons themselves or by adjacent cells or both? In situ hybridization for the ligand(s) such as Spitz might clarify this important point.

---

## [Author Response]

*1)*
Videos 1 and 2
*seem to show very different modes of movements. In*
Video 1*, it seems that vesicular transport can be seen.*
Video 2
*shows a much more diffused staining pattern. The intensity changes in this video seem to be fluctuations instead of discrete vesicular movements*.

Video 2 was only meant as an example for the method of using a heat intensity scale in order to quantify the degree of GFP fluorescence filopodia relative to background fluorescence, as shown in Figure 5. We now provide a new video of a primary neuron to show that EGFR motility in axons, branches and filopodia does indeed look very much like vesicular transport (Video 3). Furthermore, we provide evidence that EGFR-GFP puncta partially co-localize with both Rab5 and Rab11, suggesting that EGFR is present on early and recycling endosomes (Figure 5—figure supplement 2). Finally, we have succeeded in performing live imaging on EGFR-GFP in DCNs on whole brain cultures and show that it is trafficked dynamically in and out of branches and filopodia in vivo (Video 7).

*2) It is usually assumed that inhibition of endocytosis would lead to an increase in the level of receptors on the membrane.*
Figure 6
*showed the opposite. First, is it known whether the EGFR is mostly on the surface or internal? The main limitation of using a simple EGFP fusion protein is that the authors cannot distinguish between surface receptor from internal stores present in endosomes/exosomes. Second, it is entirely possible that the authors are correct and some type of transcytosis (which requires endocytosis) is sending EGFR to the filopodia. Is there any redistribution of EGFR from the cell body or*
*primary axon to the branches?*

Indeed we find EGFR is trafficked from cell bodies and main axon shafts in and out of filopodia. We now show this and in cultured neurons (Video 3) and in vivo (Figure 7—figure supplement 1). Furthermore, we provide evidence that EGFR-GFP puncta partially co-localize with both Rab5 and Rab11, suggesting that EGFR is present on early and recycling endosomes (Figure 5—figure supplement 2). It should be stated that endocytosis regulates spatial distribution of EGFR signaling in various epithelial cell types in vitro (reviewed in Ceresa, 2013). Our data demonstrate that this is also true in neurons in vivo.

*3) In*
Figure 7*, the authors use overexpressed EGFR-GFP to study the subcellular localization of EGFR. Does this construct cause there any effect on the branching phenotype? In the EGFR*^*DN*^
*background, does the EGFR-GFP rescue the phenotype? This is of course trying to test if the EGFR-GFP construct is functional*.

Both wild type untagged EGFR and tagged EGFR-GFP cause no phenotypes when overexpressed in DCNs. This is likely because, as shown in Figure 7 and its supplements, EGFR protein levels – even when highly overexpressed – are very tightly regulated and only readily detectable upon antibody staining. Furthermore, wild type EGFRs (tagged and untagged) are still regulated by ligand availability, which is not the case for EGFR^DN^ and EGFR^CA^.

When overexpressed in the eye EGFR-GFP shows similar eye defect phenotype to the overexpressed untagged wild type receptor, as stated in the original manuscript. In addition, we now show that it displays similar localization in the eye as the endogenous receptor (Figure 7—figure supplement 3).

Most importantly, as shown in the original manuscript (Figure 7—figure supplement 1) a genomic EGFR-GFP construct, tagged at *precisely* the same position as the EGFR cDNA, fully rescues the EGFR null mutant from embryonic lethality to adult viability. These data strongly argue that the GFP-tagged EGFR is fully functional.

To formally test this, we cannot ask if the UAS-EGFR-GFP rescues the dominant negative, wild type EGFR-GFP is still subject to inhibition by dominant negative EGFR. Instead, we turned to a classical assay that is used to gauge EGFR activity in vivo. Formation of the wing veins in flies requires spatially restricted EGFR activity. When wild type EGFR is expressed in the developing wing outside the normal vein domains, it causes the formation of ectopic vein tissue (e.g., Hahn et al. 2013). We find that both a control untagged wild type EGFR and EGFR-GFP cause equivalent ectopic wing vein formation (Figure 7—figure supplement 3). Altogether, our data show that a GFP tag in the C-terminus of EGFR does not compromise its activity in vivo.

*4) The critical experiment is to understand the correlation of having strong EGFR-GFP puncta on a branch and whether they will be pruned in*
Figure 7*. We did not get a clear answer whether that has been attempted or not from the text*.

To address this question in vivo (Figure 7 shows in vivo data) requires simultaneous live imaging of EGFR-GFP and DCN filopodia over a 48 hour time scale to ascertain definitively which filopodia will become the stable branches and how they behaved over a 48 hour period in relation to the level of EGFR-GFP they contain. While live imaging of DCN filopodia is currently feasible over a period of 2-4 hours (Videos 5 and 6), continuous live imaging over several days in these brain cultures is not yet possible. It should be noted that in this case one would be attempting to correlate events that take place on the scale of seconds (EGFR trafficking) with events that take place on the scale of minutes (filopodial growth and collapse), with their ultimate consequences, which take place on the scale of days (i.e., the final loss or stabilization of a branch). Furthermore, both genomic and even overexpressed EGFR-GFP are almost undetectable in brain neurons without anti-GFP antibody staining, as stated in the original manuscript. We find this to be true not only in DCNs but also in other neurons, such as LNv, also as stated in the original manuscript. For all these reasons, we had reasoned that the weakness of the signal in unstained brains precludes live imaging of EGFR-GFP puncta in DCN filopodia in vivo. This is why we performed these experiments in cultured primary neurons (Figure 5, Figure 5—figure supplement 1 and Figure 5—figure supplement 2). As stated in the original manuscript we find that small, but highly significant (p < 0.0001) differences in EGFR levels correlate with increased filopodial dynamics, both growth and retraction.

To address at least in part this important concern, we asked whether EGFR can be detected trafficking in and out of filopodia, as these filopodia move in vivo. To this end, we generated flies and developed a setup that allows high resolution imaging of EGFR-GFP movement while monitoring filopodial motion. These data (Video 7 and Figure 7—figure supplement 2) show clearly that EGFR-GFP puncta continuously enter and exit filopodia as filopodia continue to move.

Altogether, our data provide strong evidence to support the notion that dynamic asymmetries in EGFR distribution are essential for proper axonal branching and that wiping out those differences in both EGFR-DN and EGFR-CA backgrounds grinds filopodial dynamics to a virtual halt causing failure of axonal branch pruning.

*5) In*
Figure 7*, both growth and retraction of neurites are severely affected by the DN construct. The effects on the growth of the neurite are inconsistent with the lack of outgrowth phenotype in the loss-of-function analyses, which challenges the relevance of this long-term culture system*.

The effects on filopodia in EGFR loss-of-function conditions are not on the initial growth of filopodia per se (i.e., not on filopodia formation). Inhibition of EGFR stops *changes* in growth. In other words existing filopodia show reduced *net* growth and retraction after they have initially formed. The graph in Figure 7 shows the *difference* in growth and retraction rates between the genotypes and *not* the total length of filopodia. Thus, in EGFR-DN flies, initially formed filopodia no longer undergo dynamic changes, and therefore remain mostly static and develop into the excessive branches we observe in vivo. We have now changed the graph heading to reflect this and labeled the graphs as ΔGrowth and ΔRetraction to avoid the confusion we may have caused.

*6) It is puzzling that the gain-of-function EGFR causes the same phenotype as the loss-of-function. Although the authors have a way of thinking about this result, detailed analyses of the EGFR*^*CA*^
*during the developmental course is necessary to test if the loss- and gain-of-function manipulation indeed has the same effects in growth and pruning*.

In classical binary signaling paradigm, it is indeed puzzling that loss- and gain-of-function of a signal cause the same phenotype. However, what we show is that the asymmetry of EGFR signaling is essential for filopodial motion and thus we infer that any manipulation that interferes with the dynamic localization of the EGFR is likely to cause the same defect. However, as the reviewer suggests, the same final phenotype may result from different types of developmental effects. To this end, we have performed a developmental analysis of EGFR^CA^ and find that gain of EGFR function inhibits axonal branch pruning during development, mimicking the effects of EGFR loss of function (Figure 8). Again, these data provide strong evidence to support the notion that dynamic asymmetries in EGFR distribution are essential for proper axonal branching.

*7) It is unclear what is the cellular source of EGF ligand(s). The results implicate that EGF ligand(s) could act in an 'axon-autonomous' way to control branching through filopodia through contacts between filopodia of adjacent axons. However, the authors never document where EGF ligand(s) are located. Are there expressed by the axons themselves or by adjacent cells or both? In situ hybridization for the ligand(s) such as Spitz might clarify this important point*.

Our genetic (Spitz) and enhancer-reporter (Vein) data indicate that these two ligands as the likely sources of EGF activity. Spitz is likely provided by retinal axons, and not DCN axons and is thus not axon-autonomous. Therefore, in situ hybridization for the *spitz* mRNA is unlikely to reveal RNA at the axonal level. Vein is likely provided by cells adjacent to DCN axons, as shown by the Vein-LacZ reporter transgene. Furthermore, a more recent study shows that an overexpressed Spitz-GFP transgene is secreted from photoreceptor axon terminals (63), as stated in the original manuscript. To our knowledge, there are currently no Spitz antibodies that work in vivo. We were not able to find references in the recent literature (last 10 years) to such reagents. A 1998 paper (Huang et al.) does report endogenous Spitz protein expression all along the axons of all photoreceptors, consistent with our genetic evidence. We acquired the same antibody used in the 1998 study, but find that the reagent does not appear to work any longer. We were unable to detect any Spitz expression, including in positive control tissue such as the developing retina.